# Targeted high-resolution sensing of volatile organic compounds by covalent nanopore detection

Lauren E. McGivern[1,2], Zhong Hui Lim[1,2], Yizhi Yuan [1], Zonghua Bo[1], Guangqi Wu[1], Hagan Bayley [1] ✉ & Yujia Qing [1] ✉

Volatile organic compounds are choice analytes in a variety of contexts. For example, humans release over 4000 volatile organic compounds, many of which are diagnostic of life-threatening medical conditions. The analysis of combinations of a large number of potential analytes requires the application of costly, cumbersome technology. In this work, we show that covalent nanopore sensing can be used for the targeted detection of a reduced set of analytes in a mixture. In this case aldehydes, which constitute ~5% of human volatiles, can be selectively detected by using reversible thiol-aldehyde chemistry. Further, nanopore engineering permits high-resolution detection, which allows closely related aldehydes, including isomers, to be distinguished. Differential sensing of members of other chemical classes, such as mono alcohols, is also demonstrated by leveraging their enzymatic conversion into aldehydes. Our approach is compatible with the use of low-cost, portable, user-friendly diagnostic devices applicable to a wide variety of objectives, including pollutant monitoring, food and beverage testing and the quality control of pharmaceuticals, as well as disease diagnostics.

Aldehydes are ubiquitously produced through chemical processes across various biological, environmental, and industrial contexts. These include lipid peroxidation in the human body[1], incomplete combustion during wood burning[2], and Strecker degradation during beer storage[3]. Single-molecule detection of diverse aldehydes against background chemicals therefore offers wide applications.

For example, small-molecule aldehydes are found in human breath or bodily fluids, alongside a spectrum of volatile organic compounds (VOCs) generated through metabolic processes[1] or by the gut microbiome[4]. Metabolic alterations in specific conditions, such as cancers[5], gastrointestinal diseases[6], respiratory disorders[7], and viral infections[8], lead to distinct VOC profiles, which offer a non-invasive window into the body's internal biochemical processes. The current gold standard for detection of small molecules is liquid or gas chromatography-mass spectrometry (LC/GC-MS)[9], which generates a near-complete profile of all collected VOCs, but requires centralised labs that use expensive equipment and sophisticated analysis packages.

The targeted detection of aldehydes, comprising about 170 species among over 4000 VOCs of human origin[10], presents an appealing approach to reduce the complexity of small-molecule fingerprints, while retaining significant diagnostic value. Aldehyde mixtures in breath or bodily fluids commonly contain linear or branched, unsaturated or saturated carbon chains, containing from 1 to 17 carbon atoms, as well as aromatic species such as benzaldehyde and its derivatives[10]. A ratiometric fingerprint of aldehydes holds promise for disease detection and health monitoring, provided that tests can be made rapid, simple and inexpensive. For example, the ethanal:pentanal:heptanal ratio in breath shifts from 112:1:1.5 in healthy individuals to 24:1.7:1 in patients with lung cancer[11], and similarly, the hexanal:heptanal ratio in urine changes from 1.3:1 to 2.5:1[12]. For COVID

[1]Department of Chemistry, Chemistry Research Laboratory, University of Oxford, Oxford, UK. [2]These authors contributed equally: Lauren E. McGivern, Zhong Hui Lim. ✉e-mail: hagan.bayley@chem.ox.ac.uk; yujia.qing@chem.ox.ac.uk

infection, the abundance of octanal and benzaldehyde relative to each other is a hallmark of recent infection[8].

We have established single-molecule covalent sensing mediated by protein nanopores to detect analytes based on chemical reactivity[13]. The concept was first demonstrated with engineered α-hemolysin (αHL) pores[14–17] and later validated by others using alternative nanopores[18–20]. Analyte molecules form covalent bonds reversibly or irreversibly with a sensing group on the internal surface of a pore, thereby generating characteristic changes in the ionic current flowing through the pore under a transmembrane potential. When examining mixed analytes, the current signatures (e.g., the current blockade amplitude and the noise during blockades) identify analytes, while the frequency of reversible events reveals the concentration of individual analytes[21].

One general challenge is to identify sensing chemistry with suitable kinetics to ensure rapid and quantitative detection of analytes; the ideal lifetimes of covalent analyte-pore adducts and the intervals between sensing events are tens to hundreds of milliseconds, which can be reliably measured by electrical recording. This requirement has so far limited the practical application of covalent sensing. Another challenge is the rational engineering of nanopore sensors to resolve structures with minor variations, such as chain isomers that differ by the position of a methyl group. To date, this has largely relied on trial and error. A more systematic investigation is essential to guide the future development of covalent sensors.

In this work, we focus on the development of a single-molecule, real-time detection technology for rapid ratiometric profiling of aldehydes. We exploit hemithioacetal chemistry for the covalent sensing of aldehydes within a thiol-containing αHL nanopore, achieving rapid, high-resolution analysis by the formation of short-lived adducts that are identified by a machine learning algorithm. The diastereomeric adducts produce current signatures with differences that are accentuated by rational engineering of nanopores, affording information that discriminates between closely related molecular isomers. We extend the scope of our approach to mono alcohols by selectively converting them to aldehydes. Hence, we have developed a versatile means for the targeted detection of a subset of analytes present within a complex mixture of molecules.

## Results
### Covalent detection of aldehydes through reversible hemithioacetal formation

Stochastic sensing of aldehyde analytes was achieved using reversible thiol-aldehyde chemistry. To eliminate background reactions, notably imine formation and metal chelation, four mutations were introduced in wild-type (WT) αHL (i.e., AG = αHL-K8A-M113G-K131G-K147G) (Supplementary Fig. 1)[22]. A single-cysteine mutation was then introduced at position 115 (i.e., AG-T115C). Heteroheptameric nanopores containing one cysteine-bearing subunit, $(AG)_6(AG\text{-}T115C)$, were prepared and used for covalent detection (Fig. 1a, see Methods). $(AG)_6(AG\text{-}T115C)$ carried a single-channel current ($I_P$) of $-129 \pm 3$ pA at $-50$ mV (N = 80 pores) (recording buffer: 2 M KCl, 200 mM PIPES and 20 μM EDTA at pH 6.8).

The introduction of an aldehyde in the trans compartment (Fig. 1a) produced reversible current blockades, which were attributed to the formation of hemithioacetal adducts (Fig. 1b, c and Supplementary Figs. 2–10, see Supplementary Section 2). Residual currents ($I_{res}$) after hemithioacetal formation are given as percentages of the open pore current ($I_{res\%} = I_{res}/I_P \times 100\%$) (Supplementary Table 1). For example, for propanal, $I_{res\%} = 98.7 \pm 0.1\%$ (N = 3 pores, >300 events). Aldehydes exist in both hydrated and non-hydrated forms in aqueous solution[23]. The rates of hemithioacetal formation ($v_{on}$) were expressed in terms of the total aldehyde concentration and were consistent with bimolecular kinetics (i.e., for propanal, $v_{on} = k_{on}[propanal]_{tot}$, where $k_{on}$ is the observed rate constant of adduct formation). The rates of

hemithioacetal dissociation ($v_{off}$) were independent of propanal concentration, consistent with a unimolecular step (i.e., $v_{off} = k_{off}$, where $k_{off}$ is the rate constant of adduct dissociation) (Fig. 1d and Supplementary Figs. 2–9). Both the association and dissociation reactions were pH-dependent and single-channel recordings were performed at pH 6.8. At this pH value, frequent events occurred at μM-mM analyte concentrations (e.g., ~500 events within 10 min for 3 mM butanal), and the mean lifetimes of the hemithioacetal adducts were long enough to allow accurate aldehyde identification from $I_{res\%}$ values (e.g., ~130 ms for butanal).

In total, 10 different aldehydes were characterised with the $(AG)_6(AG\text{-}T115C)$ nanopore (Fig. 2a and Supplementary Table 1), ranging from straight-chain to branched-chain to aromatic aldehydes. We demonstrated single $CH_2$ resolution in distinguishing straight-chain aldehydes (Fig. 2b and Supplementary Table 1). From ethanal to octanal, each additional $CH_2$ group reduced the $I_{res\%}$ value by ~0.7%. Interestingly, our approach was able to distinguish between diastereomeric hemithioacetal adducts with opposite chirality at the Cα position for both heptanal and octanal (i.e., $\Delta I_{res\%} = 0.29 \pm 0.03\%$ for heptanal (N = 3 pores, >100 events for each diastereomer); $0.39 \pm 0.05\%$ for octanal (N = 2 pores, >40 events for each diastereomer)). We arbitrarily assigned adducts with larger $I_{res\%}$ as diastereomers A and those with smaller $I_{res\%}$ as diastereomers B (i.e., $I_{res\%,A} > I_{res\%,B}$). For shorter straight-chain aldehydes, diastereomeric adducts were not clearly separated by $(AG)_6(AG\text{-}T115C)$ with $\Delta I_{res\%} < 0.2\%$. Aromatic aldehydes produced higher $I_{res\%}$ values than saturated aliphatic aldehydes with similar masses (i.e., $I_{res\%} = 97.5 \pm 0.1\%$ for benzaldehyde (N = 3 pores, >300 events) and $96.1 \pm 0.1\%$ for hexanal (N = 3 pores, >400 events); $I_{res\%} = 96.6 \pm 0.1\%$ for phenylacetaldehyde (N = 4 pores, >400 events) and $95.7 \pm 0.1\%$ and $95.4 \pm 0.1\%$ for heptanal (N = 3 pores, >100 events for each diastereomer)). We speculate that the open-chain aldehydes extend away from the protein wall, creating larger steric blockades than the corresponding cyclic aromatic aldehydes. Branched-chain aldehydes and their straight-chain isomers produced similar $I_{res\%}$ values (i.e., $I_{res\%} = 98.1 \pm 0.1\%$ for 2-methylpropanal (N = 5 pores, >500 events) and $97.9 \pm 0.1\%$ for butanal (N = 4 pores, >400 events)), with $\Delta I_{res\%} \sim 0.2\%$ for 2-methylpropanal and butanal simultaneously recorded with the same pore (N = 1 pore, >600 events).

### Simultaneous detection of aldehydes facilitates ratiometric profiling

To enable quantitative aldehyde detection, kinetic analysis of the formation and dissociation of hemithioacetal adducts was carried out. The solubility of each aldehyde in the recording solution was measured by NMR in the presence of internal standards (5 mM calcium formate and 5 mM maleic acid). Aldehydes formed hydrates to varying extents in aqueous solutions, and the hydrated forms were unreactive towards thiols and thus not detected by single-channel recording (Fig. 1b)[24]. We determined the hydration equilibrium constants ($K_{hyd}$) by $^1H$ NMR under electrical recording conditions (Supplementary Tables 2, 3 and Supplementary Fig. 11) and derived the corrected rate constants of adduct formation ($k_{on}'$) by using the equations:

$$v_{on} = k_{on}[\text{aldehyde}]_{tot} = k_{on}'[\text{aldehyde}]_{ald} \qquad (1)$$

$$K_{hyd} = [\text{aldehyde}]_{hyd}/[\text{aldehyde}]_{ald} = [\text{aldehyde}]_{tot}/[\text{aldehyde}]_{ald} - 1 \qquad (2)$$

$$k_{on}' = k_{on} \times \left(1 + K_{hyd}\right) \qquad (3)$$

$v_{on}$, rate of hemithioacetal formation; $k_{on}$, observed rate constant of hemithioacetal formation; $k_{on}'$, corrected rate constant of hemithioacetal formation; $K_{hyd}$, hydration equilibrium constant;

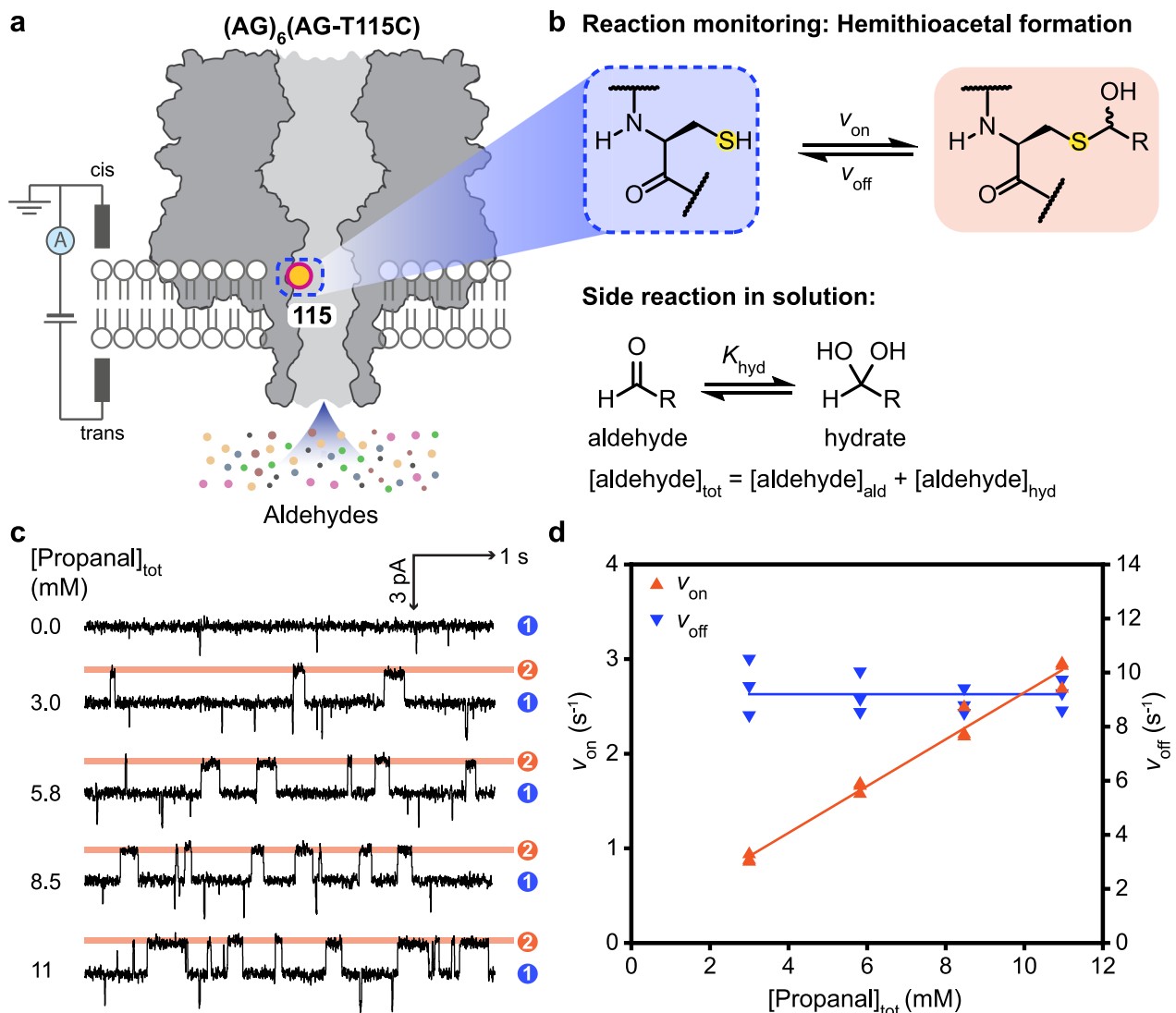

**Fig. 1 | Single-molecule covalent sensing of aldehydes. a** The engineered α-hemolysin (αHL) nanopore, (AG)$_6$(AG-T115C), bears a single cysteine at position 115 on one of the seven subunits (boxed). **b** An aldehyde molecule was detected upon reaction with the cysteine thiol to form a hemithioacetal adduct. Aldehydes are hydrated to varying extents in aqueous solutions. Concentrations of free aldehyde were calculated by using the hydration constant $K_{hyd}$ determined by $^1$H NMR. **c** Single-channel recordings at −50 mV (trans) with 0.0, 3.0, 5.8, 8.5 or 11 mM propanal (trans) in 2 M KCl, 200 mM PIPES and 20 μM EDTA at pH 6.8. Signals were low-pass filtered at 10 kHz and sampled at 50 kHz. Traces were further filtered at 100 Hz for display. Current levels corresponded to the hemithioacetal adducts formed (level 2, orange) and the unoccupied nanopore (level 1, blue). **d** Rates of adduct formation ($v_{on}$) and dissociation ($v_{off}$), recorded with individual pores, plotted against total propanal concentration showing bimolecular kinetics for the forward association step (level 1 to 2), and unimolecular kinetics for the reverse dissociation step (level 2 to 1). Source data are provided as a Source Data file.

[aldehyde]$_{tot}$, total concentration of aldehyde; [aldehyde]$_{hyd}$, concentration of aldehyde in the hydrate form; [aldehyde]$_{ald}$, concentration of free aldehyde.

For the bimolecular formation of hemithioacetal adducts, $k_{on}'$ gradually decreased as chain length increased, from ~0.5 mM$^{-1}$s$^{-1}$ for propanal to ~0.3 mM$^{-1}$s$^{-1}$ for heptanal (Fig. 2c). While the energy difference is small, these observations could reflect the steric hindrance for the thiolate to approach the carbonyl group along the Bürgi-Dunitz angle, caused by the conformationally labile alkyl chains[25]. As rates of adduct dissociation were independent of aldehyde concentration, no correction for hydration was required. For the unimolecular dissociation of hemithioacetal adducts formed with straight-chain aldehydes, values of $k_{off}$ remained within an order of magnitude, gradually decreasing, as the chain length increased, from 9.4 s$^{-1}$ for ethanal to ~5 s$^{-1}$ for hexanal, heptanal and octanal (Fig. 2c). This could be attributed to the positive inductive effect of the increasing alkyl chain length.

Simultaneous detection of multiple aldehydes with a single nanopore was demonstrated with a mixture of 7 straight-chain aldehydes (i.e., ethanal to octanal) (Fig. 3a). The consistent current signatures for individual aldehydes and the clear separation between them enabled automated assignment of events by using machine learning (see Supplementary Section 5). Almost 1000 individual events per aldehyde were collected from separate traces of each analyte (Supplementary Fig. 12). A stratified random split was performed, leaving 30% of the events as the test set. Three features were extracted to characterise each event: I$_{res\%}$, event duration, and the root-mean-squared noise of the event (see Supplementary Section 5). The random forest model achieved the highest accuracy of 98% on both the training and test sets, which was calculated as the fraction of correctly classified events in the dataset, with manual labels taken as the ground truth (Supplementary Fig. 13).

Additionally, we demonstrated ratiometric measurements of aldehydes in mixtures with pentanal and butanal, for which the detection

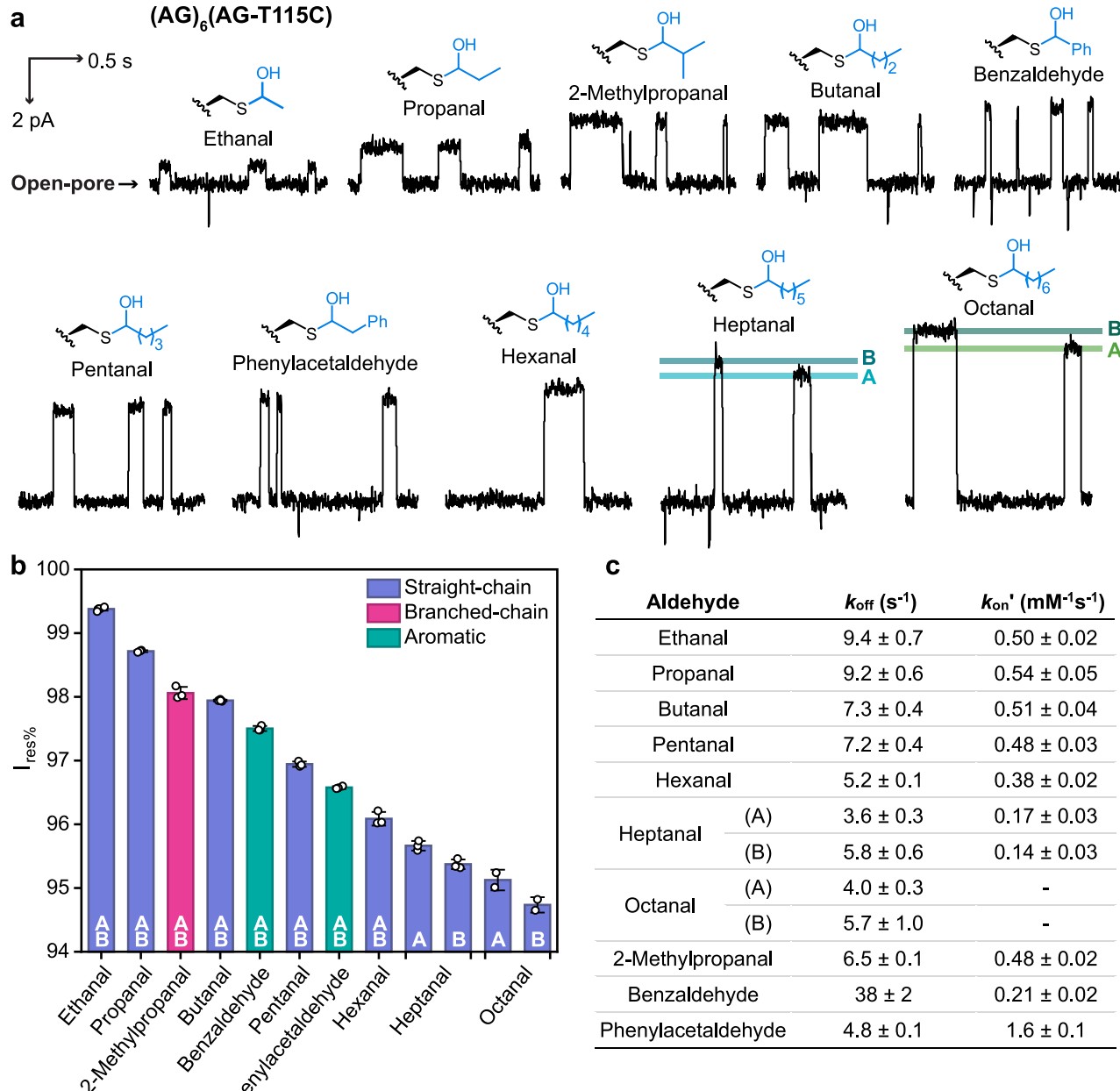

**Fig. 2 | Thiol-aldehyde chemistry within the (AG)$_6$(AG-T115C) nanopore.**
**a** Aldehydes reacted with the cysteine at position 115 to generate hemithioacetal adducts reversibly. Diastereomeric adducts were resolved for heptanal and octanal. Diastereomer A was arbitrarily assigned with a higher I$_{res\%}$ than diastereomer B. Recording conditions: −50 mV (trans) with 2 M KCl, 200 mM PIPES and 20 μM EDTA at pH 6.8. Aldehyde concentrations (trans): 5.8 mM ethanal, 5.8 mM propanal, 6.1 mM 2-methylpropanal, 5.8 mM butanal, 5.7 mM benzaldehyde, 5.5 mM pentanal, 5.5 mM phenylacetaldehyde, 2.7 mM hexanal, 1.5 mM heptanal, 1.3 mM octanal. Signals were low-pass filtered at 10 kHz and sampled at 50 kHz. Traces were further filtered at 100 Hz for display. **b** I$_{res\%}$ values for the hemithioacetal adducts. Error

bars are standard deviations of I$_{res\%}$ values determined with >3 different nanopores (Except for octanal which is with 2 different nanopores). **c** Rate constants for adduct formation corrected for hydration ($k_{on}'$) and dissociation ($k_{off}$). Errors are standard deviations across 3 different nanopores (Except for octanal which is for 2 different nanopores). Corrected rate constants for adduct formation were obtained from observed rate constants ($k_{on}$) by using the equation: $k_{on}' = k_{on} \times (1 + K_{hyd})$. Rates of adduct formation for octanal were not reported due to the poor solubility of octanal in recording solutions, precluding accurate determinations. Source data are provided as a Source Data file.

frequencies reflected their varied concentration ratios (Fig. 3b, c, see Supplementary Section 6). Within <10 minutes of single-channel recording, ratios of adduct formation events could be used to accurately determine the concentrations of pentanal and butanal (i.e., <10% difference between measured and expected concentrations) (Fig. 3c, see Supplementary Section 6). For disease diagnosis and product quality control applications, a more practical, and equally informative approach, is to directly use the ratios of adduct formation rates as fingerprints of relative concentrations of aldehydes in samples.

## Rational nanopore engineering for diastereomer and structural isomer resolution

Challenged by the similar current signatures observed with aldehyde structural isomers (e.g., butanal and 2-methylpropanal) with the (AG)$_6$(AG-T115C) nanopore, rational nanopore engineering was undertaken. We also hypothesised that current level resolution of diastereomeric hemithioacetal adducts would provide an additional layer of information to aid structural isomer resolution. To this end, we designed two additional nanopores in which the reactive cysteine

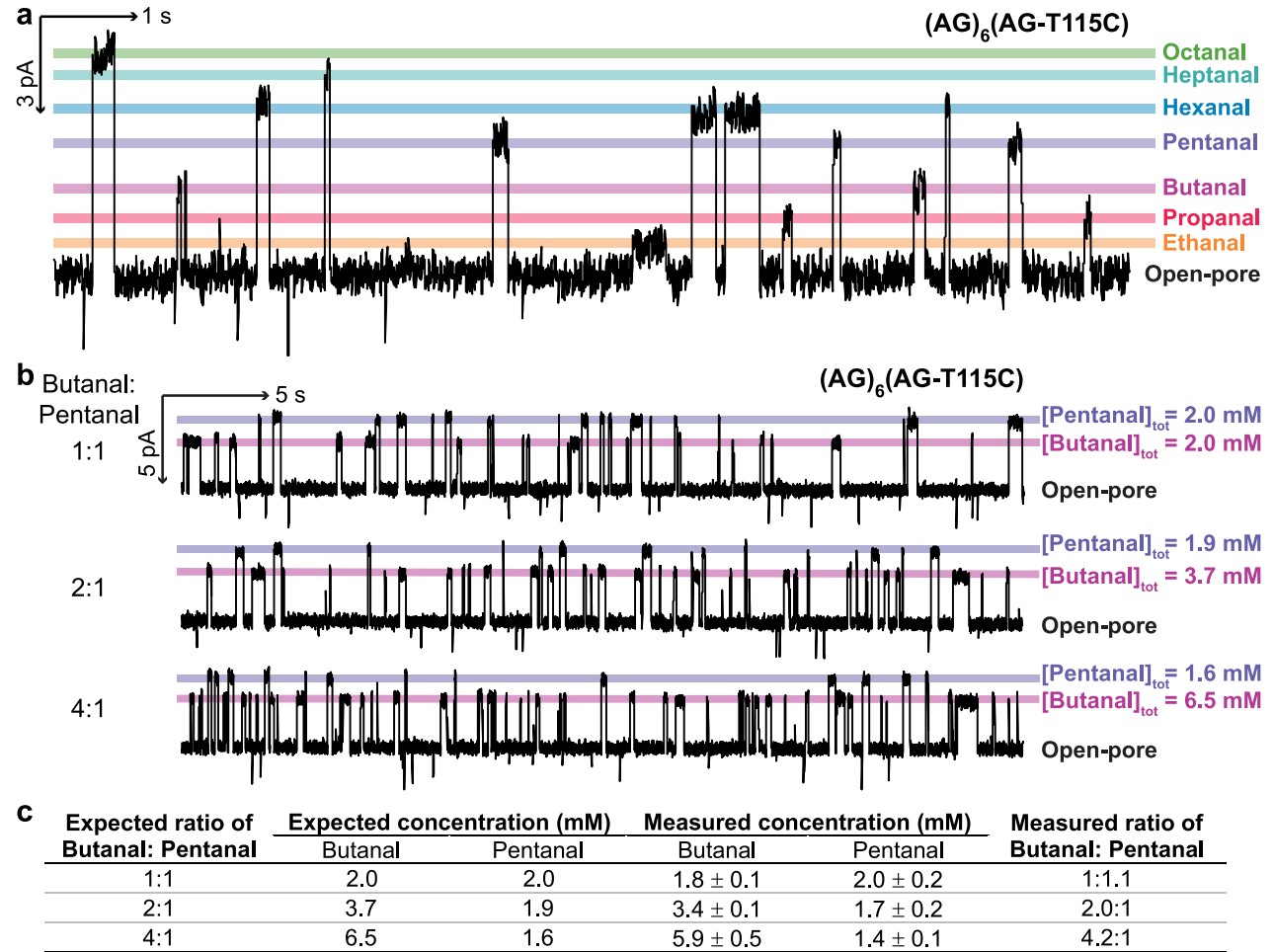

| Expected ratio of Butanal: Pentanal | Expected concentration (mM) | | Measured concentration (mM) | | Measured ratio of Butanal: Pentanal |
|---|---|---|---|---|---|
| | Butanal | Pentanal | Butanal | Pentanal | |
| 1:1 | 2.0 | 2.0 | 1.8 ± 0.1 | 2.0 ± 0.2 | 1:1.1 |
| 2:1 | 3.7 | 1.9 | 3.4 ± 0.1 | 1.7 ± 0.2 | 2.0:1 |
| 4:1 | 6.5 | 1.6 | 5.9 ± 0.5 | 1.4 ± 0.1 | 4.2:1 |

**Fig. 3 | Single-molecule profiling of aldehyde mixtures. a** Simultaneous detection of 7 straight-chain aldehydes at the single-molecule level. Total aldehyde concentrations (trans): 0.7 mM ethanal, 0.7 mM propanal, 0.7 mM butanal, 0.5 mM pentanal, 0.6 mM hexanal, 0.5 mM heptanal, and 0.3 mM octanal. **b** Ratiometric detection of mixtures of butanal and pentanal at varying ratios. Ratios of aldehydes were altered mid-experiment by incrementally replacing the analyte solution in the trans chamber with a butanal stock solution (see Supplementary Section 6).

Recording conditions: −50 mV (trans) with 2 M KCl, 200 mM PIPES and 20 μM EDTA at pH 6.8. Signals were low-pass filtered at 10 kHz and sampled at 50 kHz. Traces were further filtered at 100 Hz for display. **c** Measured total concentrations of pentanal and butanal were in good agreement with expected total concentrations (i.e., <10% difference). Errors are standard deviations of concentrations determined with 3 different nanopores. Source data are provided as a Source Data file.

residue was positioned within a narrower, and hence more sensitive, region of the nanopore β barrel (Fig. 4a). In the $(MK)_6(MK-T115C)$ nanopore (MK = αHL-K8A-K131G-K147G), methionine residues were reintroduced at position 113 to reduce the internal diameter near the sensing site (Supplementary Fig. 14). In the $(AG)_6(AG-G137C)$ nanopore, the cysteine residue was moved to position 137, which was in close proximity to the narrowest region of the nanopore bearing an AG background (Supplementary Fig. 14). Three asparagine-to-alanine mutations around the sensing site were further introduced to reduce steric hindrance and promote diastereomeric interactions of hemithioacetal adducts with the local protein environment, leading to the $(AG)_6(AG-G137C-Ala3)$ nanopore (AG-G137C-Ala3 = AG-N121A-N123A-G137C-N139A) (Fig. 4a).

The $I_{res\%}$ of 7 different straight-chain aldehydes (i.e., ethanal to octanal) were characterised with the $(MK)_6(MK-T115C)$ and $(AG)_6(AG-G137C)$ nanopores (Fig. 4b–d and Supplementary Tables 4, 5). Improved diastereomer discrimination was observed in both nanopores: diastereomeric hemithioacetal adducts from ethanal to octanal were easily separable in both nanopores: 0.3% $<\Delta I_{res\%} <1.0\%$, in the $(MK)_6(MK-T115C)$ nanopore (Fig. 4b, e and Supplementary Table 4) and 0.4% $<\Delta I_{res\%} <1.6\%$, in the $(AG)_6(AG-G137C)$ nanopore (Fig. 4d and Supplementary Table 5). Good current level separation was achieved

across all diastereomeric adducts in the $(MK)_6(MK-T115C)$ nanopore (Fig. 4e), whereas some overlap was seen in the $(AG)_6(AG-G137C)$ nanopore (e.g., $I_{res\%} = 95.9 \pm 0.1\%$ for diastereomer B of butanal and $95.8 \pm 0.1\%$ for diastereomer A of pentanal).

Current level resolution of diastereomeric hemithioacetal adducts allowed for structural isomer resolution. In the $(AG)_6(AG-G137C)$ nanopore, butanal and 2-methylpropanal could be distinguished based on diastereomers B (i.e., $\Delta I_{res\%} = 0.25 \pm 0.02\%$ for diastereomers B of butanal and 2-methylpropanal, N = 3 pores, >35 events for each aldehyde) (Fig. 4f). Further, within the $(AG)_6(AG-G137C-Ala3)$ nanopore, diastereomer pairs were better resolved for both butanal and 2-methylpropanal; distinct separation between diastereomers A enabled clear resolution of these chain isomers (i.e., $\Delta I_{res\%} = 0.42 \pm 0.03\%$ for diastereomers A of butanal and 2-methylpropanal, N = 4 pores, >60 events for each aldehyde) (Fig. 4f). In the $(MK)_6(MK-T115C)$ nanopore, butanal and 2-methylpropanal could be distinguished by using diastereomer A of 2-methylpropanal and diastereomer B for butanal (i.e., $\Delta I_{res\%} = 0.43 \pm 0.03\%$ for diastereomer B of butanal and diastereomer A of 2-methylpropanal, N = 2 pores, >50 events for each aldehyde). As a proof of concept, simultaneous detection of propanal, butanal, 2-methylpropanal and pentanal was demonstrated in the $(MK)_6(MK-T115C)$ nanopore (Fig. 4g). As the

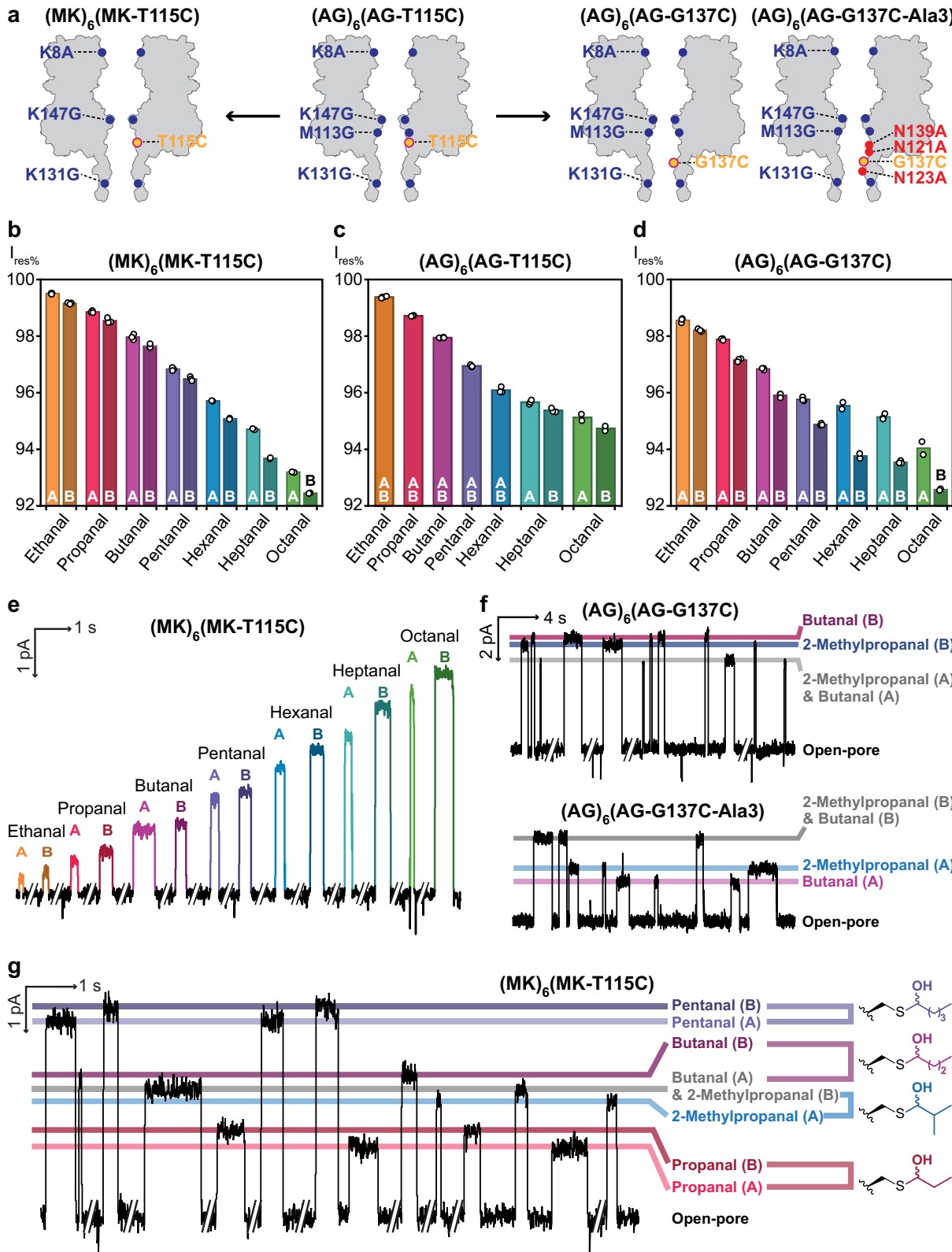

current level blockades for pentanal and propanal do not overlap with those of butanal and 2-methylpropanal, all four aldehydes could be differentiated from $I_{res\%}$ alone.

### Enzyme-assisted differential sensing of alcohols and aldehydes

Nanopore covalent sensing is inherently selective for a targeted class of analytes. For example, in a mixture of alcohols (1-pentanol and 1-hexanol) and aldehydes (propanal and butanal), only the aldehydes produced events within the $(AG)_6(AG\text{-}T115C)$ nanopore sensor (Fig. 5, Left). After treatment with an engineered alcohol oxidase[26], additional aldehyde species—pentanal and hexanal—were identified (Fig. 5, Right), revealing the presence of alcohols in the original sample. This simple functional group conversion step, coupled to nanopore sensing, therefore allowed for differential single-molecule detection of

**Fig. 4 | Diastereomer discrimination in engineered nanopores facilitates structural isomer resolution. a** Rational engineering of nanopores to enhance diastereomer discrimination through two parallel strategies: Left: narrowing the pore diameter around the sensing group (yellow); Right: moving the sensing group to the narrowest internal site. Mutations are highlighted relative to the WT background in all subunits (blue) or in the cysteine-bearing subunit (red). **b** Within the $(MK)_6(MK\text{-}T115C)$ nanopore, diastereomeric adducts were resolved for all straight-chain aldehydes tested. Diastereomer A was arbitrarily assigned with a higher $I_{res\%}$ than diastereomer B. **c** Within the $(AG)_6(AG\text{-}T115C)$ nanopore, diastereomeric adducts were resolved for only heptanal and octanal. **d** Within the $(AG)_6(AG\text{-}G137C)$ nanopore, diastereomeric adducts were resolved for all straight-chain aldehydes

tested. **e** Good current level separation was achieved across all diastereomeric adducts in the $(MK)_6(MK\text{-}T115C)$ nanopore. **f** A pair of chain isomers, butanal and 2-methylpropanal, were distinguished by diastereomers B in the $(AG)_6(AG\text{-}G137C)$ nanopore or by diastereomers A in the $(AG)_6(AG\text{-}G137C\text{-}Ala3)$ nanopore. **g** Propanal, butanal, 2-methylpropanal, and pentanal were detected and distinguished within a single $(MK)_6(MK\text{-}T115C)$ pore. Butanal and 2-methylpropanal were distinguished by the diastereomer A of 2-methylpropanal and the diastereomer B of butanal. Recording conditions: −50 mV (trans) with 2 M KCl, 200 mM PIPES and 20 μM EDTA at pH 6.8. Signals were low-pass filtered at 10 kHz and sampled at 50 kHz. Traces were further filtered at 50 Hz for display. Source data are provided as a Source Data file.

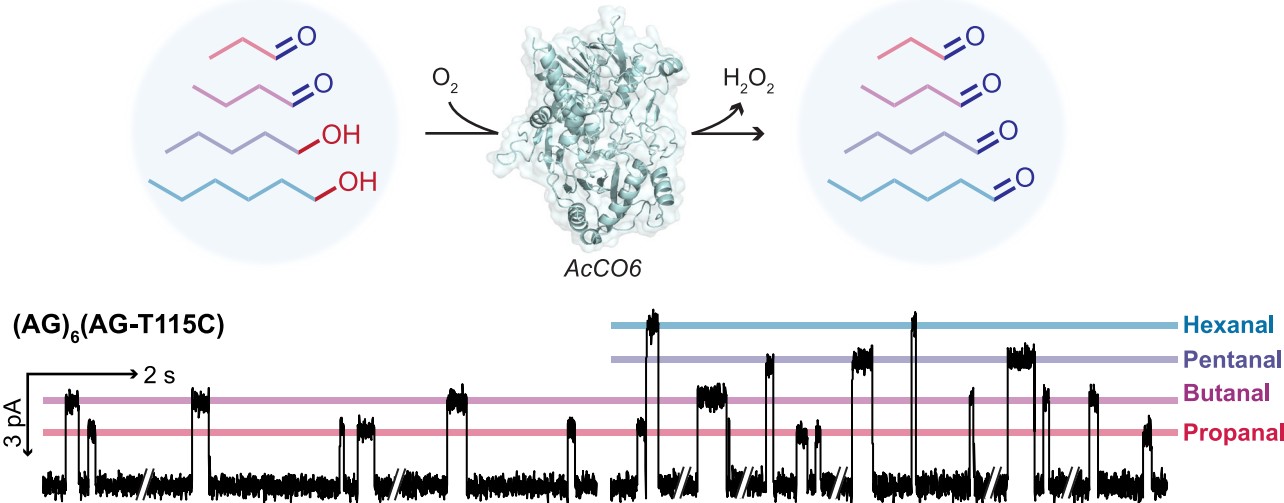

**Fig. 5 | Enzyme-facilitated indirect alcohol detection.** The $(AG)_6(AG\text{-}T115C)$ nanopore selectively detected aldehydes in an aldehyde-alcohol mixture (Left). After treatment with an engineered alcohol oxidase (AcCO6), additional aldehyde species−pentanal and hexanal−were observed (Right), which indicated the presence of the corresponding alcohols in the original sample. Recording conditions: −50 mV (trans) with 2 M KCl, 190 mM PIPES and 20 μM EDTA at pH 6.8. Signals were low-pass filtered at 10 kHz and sampled at 50 kHz. Traces were further filtered at 100 Hz for display.

alcohols and aldehydes in a mixture−a powerful strategy worth further development.

## Discussion

The present work exploited thiol-aldehyde chemistry, a previously unexplored dynamic covalent chemistry within protein nanopores, to target the sensing of aldehydes, a key class of chemicals ubiquitous in daily life. We achieved single-molecule identification of 10 straight-chain, branched-chain, and aromatic aldehydes, which represent potential biomarkers for lung cancer[5], Crohn's disease[6], SARS-CoV-2[8], as well as being atmospheric pollutants[2], and beverage impurities[3]. Our approach resolved straight-chain aldehydes that differed by single $CH_2$ groups, members of hemithioacetal diastereomeric pairs, and structural isomers−challenges often faced by conventional small-molecule detection methods. Compared to traditional LC/GC-MS methods, pre-derivatisation of aldehydes (e.g., with hydrazine-based chromophores) was not required[27]. Additionally, we demonstrated rational engineering of the protein environment surrounding the covalent sensing site, achieving improved signal separation for diastereomeric adducts by promoting interactions with the pore interior. Rational engineering of the sensing region thus holds promise for improving small-molecule covalent detection with a given nanopore scaffold.

As a proof of concept, we established ratiometric profiles of mixed aldehydes at mM concentrations with >400 events collected in 10 min with a single nanopore. Given that aldehydic VOCs ranging from propanal to nonanal are exhaled at 3-300 pM concentrations[28], in

a commercialised device, pre-concentration steps would be coupled to nanopore detection. For example, 3-300 μM concentrations of aldehydes can theoretically be obtained by pre-concentrating 5 L of breath volatiles into 5 μL of solvent by using solid-phase microextraction (SPME)[29,30]. Detection of analytes in nanolitre droplets (i.e., 300 nL) was previously demonstrated in a microfluidics set-up, and could also be integrated here[31]. Further, selective enrichment of aldehydes has been achieved using SPME devices with fibre coatings selective for aldehydes[28,32,33]−this would simultaneously eliminate potentially interfering chemical classes (e.g., disulfides) that might hinder disease diagnosis. For longer chain aldehydes with poor solubilities (i.e., <1 μM solubilities), the concentration of VOCs into solvents such as water/ethanol solutions might be considered, although this would require alternative bilayer systems with greater tolerance to organic solvents. Improved detection frequency might also be achieved by using nanopore engineering, e.g., mutants with multiple covalent sensing sites[14] or mutagenesis of proximate residues to improve cysteine thiol reactivity[34]. In addition, nanopore sensing devices containing arrays of pores would enhance throughput (e.g., the MinION device contains up to 512 active pores per flow cell)[35]. Based on our findings, we envision an accessible nanopore platform for rapid aldehyde detection, paving the way for applications in disease diagnosis, environmental monitoring, and the quality control of food and beverages.

Moving forward, the detection of other biologically relevant chemical classes can be explored by capitalising on the aldehyde-sensing system developed here. As central metabolites, aldehydes are generated by a wide range of enzymes from various functional groups,

including carboxylic acids, primary alcohols, and primary amines[36]. Many of these convertible chemical classes are also potential disease biomarkers. For example, 1-pentanol, detected as 1-pentanal in this work, has been found in the exhaled breath of lung cancer patients but not healthy individuals[5]. While we have demonstrated mono alcohol sensing after enzymatic conversion to aldehydes, the detection of other chemical classes will require the application of means for the direct chemical conversion to reactive molecules or the identification of suitable enzymes for conversion, particularly with respect to substrate scope and catalytic efficiency. In the long term, to leverage thiol-aldehyde sensing chemistry, we envision a versatile sensing workflow that employs a suite of reagents for conversion to aldehydes, followed by rapid single-molecule detection in low-cost, portable, user-friendly devices.

## Methods

### Preparation of plasmids

Construction of plasmids encoding mutant αHL monomers, together with the primers required, are described in the Supplementary Information. Plasmids are available from the corresponding authors upon request.

### Preparation of nanopore sensors

Nanopore heteroheptamers were prepared according to the procedure below, which is a modification of a method previously reported[37].

αHL monomers were prepared with an *E. coli* in vitro transcription and translation (IVTT) system (*E. coli* T7 S30 Extract System for Circular DNA, Cat #L1130, Promega). Prior to use, the T7 S30 extract provided in the kit was treated with 1 μL rifampicin (1 μg/mL, final concentration) to suppress transcription by *E. coli* RNA polymerase. A standard reaction comprised: DNA plasmid mixture (< 4 μg, plasmids encoding cysteine-free and cysteine-containing subunits were in an 8:1 ratio), amino acid mixture without methionine (5 μL, as supplied in the kit), S30 premix without amino acids (20 μL, as supplied in the kit), [$^{35}$S] methionine (2 μL, 1200 Ci/mmol, 15 mCi/mL, MP Biomedicals), rabbit red blood cell membranes (2 μL, ~1 mg protein/mL), and T7 S30 extract for circular DNA (15 μL, as supplied in the kit). The reaction mixture was incubated at 37 °C for 2 h.

Next, MBSA buffer (1 mL; 3-morpholinopropane-1-sulfonic acid (10 mM), NaCl (150 mM), bovine serum albumin (1 mg/mL), pH 7.4) was added to the reaction mixture. The mixture was centrifuged at 25,000 x g for 10 min at 4 °C. The supernatant was then removed, and the pellet solubilised at room temperature with 2× Laemmli sample buffer (25 μL) which contains 10% 2-mercaptoethanol. Before loading the samples, a 5.5% SDS/PAGE gel was pre-run with 1× Tris-Glycine SDS running buffer containing dithiothreitol (2 mM) and sodium thioglycolate (1 mM). The resuspended pellets were then loaded onto the gel and electrophoresed at +70 V overnight (13 h).

αHL heptamers containing different numbers of mutant subunits were separated in the gel based on their electrophoretic mobilities which were determined by the number of octa-aspartate (D8) tails present (i.e., each cysteine-bearing mutant subunit contained a D8 tail). Hence, the top band corresponded to homoheptamers bearing no octa-aspartate tail (i.e., (AG)$_7$ or (MK)$_7$), the second band corresponded to heteroheptamers bearing a single octa-aspartate tail (i.e., (mutant-D8)$_1$(AG)$_6$ or (mutant-D8)$_1$(MK)$_6$) and so on, with consecutive bands having an aspartate-tail-free subunit replaced with a mutant subunit bearing an octa-aspartate tail. In this work, the second band from the top contained the desired protein pore with a single cysteine residue.

To extract the protein pores, the gel was dried under vacuum onto Whatman 3MM filter paper for 5 h at 50 °C. The dried gel was then exposed to photographic film (Kodak Bio Max MR autoradiography film) for 8 h and the developed film was used to locate the target protein bands in the gel. The desired protein bands were excised and rehydrated in TE buffer (300 μL; Tris·HCl (10 mM), DTT (0.5 mM), EDTA (1 mM), pH 8.0) for 1 h at room temperature. The paper was then removed, and the gel was crushed with a plastic pestle. The resulting suspension was filtered through a 0.2 μm hydrophilic membrane filter (Proteus Mini Clarification Spin Column, Generon). The filtrate was stored in 10 μL aliquots at −80 °C.

### Single-channel electrical recordings

Single-channel recordings were carried out in a planar bilayer apparatus as previously described[38]. A single αHL pore was allowed to insert into the bilayer. Aldehyde substrates were introduced from the trans compartment. Experiments were conducted using recording buffer containing 2 M KCl, 200 mM PIPES and 20 μM EDTA titrated to pH 6.8. Aldehyde solutions were prepared with recording buffer. Single-channel recordings were conducted with a coverslip placed atop the recording chamber.

Ionic currents were recorded by using a patch clamp amplifier (Axopatch 200B, Molecular Devices), and filtered with a low-pass Bessel filter (80 dB/decade) with a corner frequency of 10 kHz. Signals were digitised with a Digidata 1320 A digitizer (Molecular Devices) at an acquisition frequency of 50 kHz. The current traces were processed with Clampfit 10.7 (Molecular Devices). Current traces were idealised by using Clampfit 10.7 (Molecular Devices). The idealised data were analysed with QuB 2.0 software (www.qub.buffalo.edu)[39]. Dwell time analysis and rate constant determinations were performed by using the maximum interval likelihood algorithm of QuB[40].

### *Arthrobacter cholorphenolicus* choline oxidase (AcCO6) preparation

The plasmid pET28a-AcCO6 encoding an engineered *Arthrobacter cholorphenolicus* choline oxidase (AcCO6) was a kind gift from Nicholas J. Turner[26]. The plasmid was transformed in *E. coli* BL21(DE3) competent cells for expression, which were cultured in LB media (10 mL) containing 50 μg/mL kanamycin at 37 °C, 220 RPM overnight. The overnight culture was transferred to autoinduction media (1 L, LB based, Formedium, #AIMLB0110) containing 50 μg/mL kanamycin and grown at 220 RPM for 16 hours at 20 °C. Cells were harvested by centrifugation at 4000 x g for 20 min at 4 °C. The pellet was resuspended in 50 mL pre-chilled Buffer A (100 mM KPi pH 7.8, 300 mM KCl, 20 mM imidazole) supplemented with 250 U nucleases (Pierce™ Universal Nuclease for Cell Lysis, #88700). Cells were lysed by sonication (10 s pulse, 10 s pause, 3 min duration) on ice, followed by centrifugation at 40,000 x g for 30 min at 4 °C. The supernatant was filtered through a 0.45 μm syringe filter and applied to a 5 mL HisTrap™ FF column (Buffer A: 100 mM KPi pH 7.8, 300 mM KCl, 20 mM imidazole; Buffer B: 100 mM KPi pH 7.8, 300 mM KCl, 1000 mM imidazole). Fractions containing His-tagged AcCO6 were pooled and dialysed against a buffer containing 100 mM KPi pH 7.8 at 4 °C overnight. The His-tagged AcCO6 was concentrated, aliquoted, flash frozen with liquid N$_2$, and stored at −80 °C.

### Differential sensing of aldehydes and alcohols

The control aldehyde-alcohol mixture was prepared as follows: propanal, butanal, 1-pentanol, 1-hexanol (5 mM each) were dissolved in oxygen-saturated KPi buffer (100 mM, pH 7.8). The treated mixture was prepared as follows: AcCO6 (0.1 mg/mL for each mM of alcohol) was added to the control aldehyde-alcohol mixture and shaken overnight at 300 RPM at 30 °C. After incubation, both the control and treated mixtures were adjusted to pH 6.8, and diluted 4-fold with buffer (2.66 M KCl, 200 mM PIPES at pH 6.8, 20 μM EDTA) (i.e., final concentration of 2 M KCl).

Either the control (100 μL) or treated mixture (100 μL) was introduced into the trans chamber. As the control mixture was not treated with AcCO6, only propanal and butanal could be detected.

Single-channel recordings with the treated mixture revealed additional events with conductance levels corresponding to hemithioacetal adducts derived from pentanal and hexanal. This confirmed the oxidation of alcohols to aldehydes after AcCO6 treatment.

**Note added in proof**: During the production of this paper, we reported complementary findings[41] on the diastereoselectivity of reactions between a nanoreactor cysteine and aromatic aldehydes. These results further confirm that protein nanopores can discriminate hemithioacetal diastereomers at the single-molecule level.

## Data availability
Data supporting the findings of this study are shown in the main text, Supplementary Information and Source data. Data needed to reproduce the machine learning results are available in the repository: https://doi.org/10.5281/zenodo.16949339. Source data are provided with this paper.

## Code availability
Codes needed to reproduce the machine learning results are available in the repository: https://doi.org/10.5281/zenodo.16949339.

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

## Acknowledgements

This research was supported by the Bill & Melinda Gates Foundation, a European Research Council Advanced Grant (SYNTISU), and a European Research Council Starting Grant (NANOPRO) under the UK Research and Innovation (UKRI) Guarantee (EP/Z000351/1). Y.Y. is supported by an EPSRC Postdoctoral Fellowship (UKRI939). G.W. thanks the support of UKRI for the Horizon Europe Guarantee funding of a Marie Skłodowska-Curie Actions Postdoctoral Fellowship (EP/Z001366/1). We thank Prof. Nicholas J. Turner from the University of Manchester for providing the AcCO6-encoding plasmid. We thank Steven Barry at the Physical and Theoretical Chemistry Laboratory, University of Oxford, for fabricating the recording chambers.

## Author contributions

H.B. and Y.Q. conceived and supervised the project. L.E.M. and Z.H.L. prepared the proteins and conducted the single-channel recording experiments. Z.B. prepared the script for machine learning and performed the molecular dynamics simulations. Y.Y. prepared the AcCO6 enzyme. G.W. carried out parallel sensing experiments. L.E.M., Z.H.L., Z.B. and Y.Q. performed the data analysis. L.E.M., Z.H.L., Y.Y., Z.B., H.B. and Y.Q. wrote the manuscript.

## Competing interests

H.B. is the founder of, a consultant for and a shareholder of Oxford Nanopore Technologies, a company engaged in the development of nanopore sensing and sequencing technologies. L.E.M., Z.H.L., H.B. and Y.Q. have filed patents describing the engineered nanopores and their applications in small-molecule covalent sensing. Z.B. and Y.Y. are listed as contributors to the IP. The remaining authors declare no competing interests.
