## [Transparent Peer Review file · Nature Communications]

Targeted high-resolution sensing of volatile organic compounds by covalent nanopore detection

Corresponding Author: Professor Yujia Qing

Version 0:

Reviewer comments:

Reviewer #1

(Remarks to the Author)

The identification of individual aldehyde molecules, which can serve as biomarkers, has been achieved through machine learning applied to ion current–time waveforms from biological nanopores. These nanopores are chemically programmed via their amino acid sequences. Notably, aldehyde isomers have been identified with high accuracy at the single-molecule level, and even individual CH₂ units within aldehydes have been distinguished. A particularly notable finding is the comparison of the chemical reaction rates between aldehydes and thiols in bulk versus single-molecule settings. This study has the potential to enable high-precision single-molecule aldehyde sensing and to open new avenues for studying single-molecule chemical reactions within localized biomolecular environments. After addressing the following concerns, I believe that this paper is suitable for publication in this journal.

The introduction focuses on quantitative analysis at the single-molecule level. However, the experimental data do not directly support such an analysis. While machine learning is applied to individual ion current–time waveforms, and the number of waveforms classified as each aldehyde is determined, this study does not explicitly use this count for quantitative analysis. This raises an important question: Can quantitative analysis be performed simply by counting the number of ion current–time waveforms classified as each aldehyde? If so, this aspect should be clarified and further explored.

(Remarks on code availability)

I used Jupyter Notebook to check that the code worked.

Reviewer #2

(Remarks to the Author)

In the manuscript "Targeted, high-resolution sensing of volatile organic compounds by covalent nanopore detection", Lauren E M. and co-authors presents a groundbreaking approach to the sensing of aldehyde compounds through a covalent nanopore-based sensing platform, offering high-resolution detection with single-molecule manner. However, I think this method might lack of meaningful application and practical significance-then it might not worth publication on Nature Communications. Recent works, such as those reported in Trends in Analytical Chemistry (171, 117548, 2024) and Microchemical Journal (199, 110051, 2024), have demonstrated the analysis of aldehydes in volatile organic compounds (VOCs) and the environment with a limit of detection (LOD) as low as ~0.1 nM. This sensitivity is significantly higher than the current nanopore methods, which operate at the millimolar level. While the authors highlight the high-resolution sensing capabilities of their nanopore method for aldehyde isomers, such precision may currently hold limited practical value, especially when considering the cost-effectiveness and simplicity of alternative approaches, which is much more sensitive than the current nanopore methods (millimolar level). Although the author considered the nanopore methods featured high-resolution sensing for aldehydes isomers, such precision holds few practical values at the present concentration even in a single-molecule manner, particularly when considering the cost-effectiveness and simplicity of alternative approaches. Furthermore, the authors do not provide sufficient discussion on the scientific advancements and innovations that distinguish their work from previous efforts with similar goals. It would be helpful to understand the unique contributions of this study and why it warrants publication in a high-impact journal like Nature Communications.

Additionally, the title mentions "targeted" sensing, but the authors do not clearly explain what this term entails. Clarification on the targeting mechanism and its specificity would be beneficial.

In the authors' previous work, it was demonstrated that many types of small molecules can react with the T115C site inside the nanopore. However, the current study lacks an investigation into potential interference from other compounds, leading to insufficient discussion regarding the method's selectivity. A comprehensive analysis of potential interferences and strategies to mitigate them would strengthen the manuscript.

Moreover, the fundamental for the formation of hemithioacetal underlying this work are not supported by sufficient experimental evidence, raising questions about the reliability and validity of the proposed approach. Mass or MRI evidences were suggested for the formation of hemithioacetal between the AG-T115C and aldehydes substrates in single molecular experimental conditions. The authors conducted rigorous experiments to rule out changes in the substrate concentration caused by aldehyde hydrolysis. In that case, can the formation of the hemithioacetal only occur via nucleophilic addition reaction? Could the nucleophilic substitution of the hydrolysis products also be a possible reaction pathway? The likelihood of this pathway might be related to the pH values.

The assertion that events A and B stem from reaction-formed enantiomers of the same aldehyde lacks robust experimental evidence. To strengthen this claim, the authors could provide definitive evidence linking the A/B event to the specific R/S configuration of the aldehyde enantiomers. This could open up an exciting new direction for further research and help solidify the understanding of the underlying mechanisms. Additionally, in Figure 4d, the I_{res} % values for pentanal and hexanal display unexpected increases, which remain inadequately explained in the manuscript. Clarifying this anomaly would greatly enhance the comprehensiveness of the study.

In the third paragraph of the introduction part, the author elaborated on the significance of rational detection of various VOC. However, this section lacks a direct connection with the subsequent experiments. This disconnection not only confuses the readers but undermines the logical coherence of this part. If these measurement methods are not further validated or applied in the experiments, then dedicating extensive space to discussing their significance in the preface appears redundant and inappropriate.

And some small questions:

The chemical reaction equation in figure 1 seems confusion.

The figure 2 lacks the current of the open-pore state.

The koff for Benzaldehyde with the nanopore was much higher than other aldehydes, while its kon is reasonable. Further interpretation is suggested of this kinetics data.

The koff of isomer B was generally higher than that of its isomer A. If the two signals indeed arise from the chirality induced post-addition, it raises a compelling question as to why the dissociation rates of the addition products with different chirality exhibit such significant disparities.

(Remarks on code availability)

Reviewer #3

(Remarks to the Author)

With great interest I read the new manuscript by H. Bayley and coworkers. The team is among the leading one in the area of nanopore sensing having an enormous expertise in using hemolysine as molecular sensor. Here they presents novel ways to detect small aldehydes by nanopores. Sensing is achieved by blocking the ion current in a specific manner to allow discrimination of a larger number of different aldehydes. For this the team introduced a cysteine next to the constriction zone. If an aldehyde passes the cysteine they will react and bind temporarily and modulate eventually the ion current. Further enhancement in selectivity was done by additional mutations giving a certain cooperativity. Theoretically this is obvious in the nanopore field, however, finding an appropriate binding site and causing specific ion current modulation is a real challenge. Able to discriminate different aldehydes is a rather unique finding.

I think this is a beautiful example for enzyme engineering in a rational manner with single molecule resolution. With the current goal of bringing nanopore detection to the market this approach would be a possible further application.

However, I see a few bottlenecks not mentioned in the otherwise very clearly written introduction: how to bring volatile molecules into an aqueous solution knowing that their solubility is very different or even very low for various aldehydes (some have even very low solubility). This is particular important as the ratio seems to matter often. Maybe using water/ethanol solutions and solid state nanopores? To my knowledge nobody has tried that).

The second bottleneck is the different reaction rate already mentioned in the manuscript.

Would adding higher PEG concentration help to dehydrate the aldehydes?

You use lipid membranes, what about partitioning of hydrophobic aldehydes into the membrane reducing the concentration in a specific manner?

The work is complete and fully described. To reproduce is technically a challenge but feasible for competing groups. The manuscript is sound and at the current edge in the nanopore field.

Overall this work is a significant step and I recommend publication.

(Remarks on code availability)

Hi, yes I ve seen it. Unfortunately I have to confess that I didn t look at it, this requires a specialist in ML, shall I ask someone (U. Kleinekathöfer, or M. Ceccarelli...)

On the other hand I find it useful so if one wishes, you can redo it. This is very crucial, I'm often lost as ML gives data and I'm not sure if it's true.

Let me know

Version 1:

Reviewer comments:

Reviewer #1

(Remarks to the Author)

The authors fully improved the manuscript and addressed my questions satisfactorily. I think that the manuscript is ready for publication in Nature Communication.

(Remarks on code availability)

I confirmed that the code used in this paper is the well-established open-source scikit-learn.

Reviewer #2

(Remarks to the Author)

While the authors have made certain revisions, the fundamental concerns regarding the scientific and practical validity of the proposed method remain unresolved. The core claims are not sufficiently supported by data, and the potential for practical application is, at best, remote. I do not recommend this work for publication in its current form.

Firstly, the claim that this approach could be applied to point-of-care testing (POCT) for VOCs remains speculative. Although the authors reference various strategies such as preconcentration, multi-channel detection, and nanopore engineering, these are drawn from unrelated contexts and are not convincingly integrated into their own method. The manuscript fails to provide any compelling evidence that the proposed system is truly low-cost, rapid, or user-friendly. In fact, critical barriers—including chip degradation in the MinION platform and the extensive computational resources required for nanopore signal processing—are entirely overlooked. Without addressing these limitations, the method cannot be considered viable for practical applications.

Secondly, the authors continue to ignore the crucial role of the protein microenvironment in modulating cysteine reactivity, a fact well established in the literature and even acknowledged in their prior work. The supporting data—primarily derived from simplified chemical models—bear little relevance to the biological setting in which the claimed sensing would occur.

Moreover, there is still no investigation into whether the monomers forming the nanopore might themselves participate in the reaction, calling into question the biological plausibility and specificity of the approach.

Thirdly, the mechanistic assumptions presented remain weak. The formation of hemiacetals is known to proceed via both acid- and base-catalyzed pathways, yet the authors make no attempt to experimentally assess pH dependence. Instead, they selectively cite literature to justify a nucleophilic addition mechanism, without conducting even a minimal set of control experiments. This lack of mechanistic rigor severely undermines the reliability of the proposed sensing strategy, particularly if it is to be applied in complex or variable environments.

(Remarks on code availability)

Reviewer #3

(Remarks to the Author)

I read the ms. as well as the reply. As stated in my first report the manuscript contains a novel approach which will raise sufficient interest in the community.

The authors improved the ms. and it can be published as it is

(Remarks on code availability)

Version 2:

Reviewer comments:

Reviewer #1

(Remarks to the Author)

[Editorial note: Please note that Reviewer #1 assessed authors' responses to Reviewer #2 and considers them addressed]

RESPONSE TO REVIEWERS

Reviewer #1 (Remarks to the Author):

The identification of individual aldehyde molecules, which can serve as biomarkers, has been achieved through machine learning applied to ion current–time waveforms from biological nanopores. These nanopores are chemically programmed via their amino acid sequences. Notably, aldehyde isomers have been identified with high accuracy at the single-molecule level, and even individual CH₂ units within aldehydes have been distinguished. A particularly notable finding is the comparison of the chemical reaction rates between aldehydes and thiols in bulk versus single-molecule settings. This study has the potential to enable high-precision single-molecule aldehyde sensing and to open new avenues for studying single-molecule chemical reactions within localized biomolecular environments. After addressing the following concerns, I believe that this paper is suitable for publication in this journal.

We thank the reviewer for these positive comments.

C1: The introduction focuses on quantitative analysis at the single-molecule level. However, the experimental data do not directly support such an analysis. While machine learning is applied to individual ion current–time waveforms, and the number of waveforms classified as each aldehyde is determined, this study does not explicitly use this count for quantitative analysis. This raises an important question: Can quantitative analysis be performed simply by counting the number of ion current–time waveforms classified as each aldehyde? If so, this aspect should be clarified and further explored.

A1: We clarify our approach as follows:

1. The rate of adduct formation (v_{on}) is concentration-dependent (i.e., higher v_{on} is obtained with higher concentrations of aldehydes, reflected as a shorter mean waiting time before adduct formation), and varies between aldehydes based on their intrinsic reactivity. Reactivity is quantified by the aldehyde's rate constant of adduct formation ($k_{on} = v_{on} / [\text{aldehyde}]$).
2. If proper calibration of k_{on} is done for all aldehydes (see Supplementary Section 2), absolute concentrations of each aldehyde can be obtained by measuring the corresponding v_{on} for each aldehyde (see Experimental methods). This is now demonstrated in the manuscript (Fig. 3b,c).
3. A more practical, and equally informative approach, is to directly use the ratio of v_{on} as fingerprints of relative concentrations of aldehydes in biological samples. For example, for aldehydes A and B, v_{on}^A / v_{on}^B is linearly proportional to $[A] / [B]$.

These points are now discussed and exemplified in the manuscript (Page 10, lines 3-12; Fig. 3), and comprehensively detailed in Supplementary Section 6. Key results are discussed below:

Concentrations of pentanal and butanal in mixtures of different ratios were calculated (Table R1, now included as Fig. 3c). Measured concentrations were in good agreement with expected concentrations (i.e., <10% difference between measured and expected concentrations).

Table R1: Measured concentrations and ratios of aldehydes in pentanal/butanal mixtures.
[a,b]

Expected ratio of Butanal: Pentanal	Expected concentration (mM)		Measured concentration (mM)		Measured ratio of Butanal: Pentanal
	Butanal	Pentanal	Butanal	Pentanal	
1:1	2	2	1.9 ± 0.2	2.1 ± 0.3	1:1.1
2:1	3.7	1.9	3.6 ± 0.1	1.8 ± 0.2	2.0:1
4:1	6.5	1.6	6.1 ± 0.6	1.5 ± 0.1	4.2:1

[a] Errors are standard deviations of concentrations for N = 3 nanopores.

[b] Ratios and concentrations were obtained from <10-minute single-channel recordings.

The time needed for accurate ratiometric measurements is dependent on the total concentration of the aldehydes present, as well as the ratio of aldehydes. We found that approximately 100 events per aldehyde was sufficient to provide accurate measurements (i.e., <10% difference between measured and expected concentrations). This was achieved with a recording time of <10 minutes for aldehyde concentrations above 1.5 mM. However, a longer time would be needed for aldehyde mixtures at lower concentrations, or if the ratio of aldehydes were higher (e.g., 10:1). These issues could be addressed by improving the limits of detection as discussed in the Conclusions, including:

1. Nanopore engineering, e.g., nanopore mutants engineered with more than one covalent sensing site [Wu, JACS, 2008] or site-directed mutagenesis of proximate residues to increase cysteine thiol reactivity [Bo, ACIE, 2023].
2. Nanopore sensing devices containing arrays of pores, e.g., the MinION device contains up to 512 active pores per flow cell [Dreamer, Nat. Biotechnol., 2016]).

References:

- [1] Wu, H.-C. & Bayley, H. Single-Molecule Detection of Nitrogen Mustards by Covalent Reaction within a Protein Nanopore. *J. Am. Chem. Soc.* **130**, 6813-6819 (2008). DOI: 10.1021/ja8004607.
- [2] Bo, Z., Lim, Z. H., Duarte, F., Bayley, H. & Qing, Y. Mobile Molecules: Reactivity Profiling Guides Faster Movement on a Cysteine Track. *Angew. Chem. Int. Ed.* **62**, e2023008 (2023). DOI: 10.1002/anie.202300890.
- [3] Dreamer, D., Akeson, M. & Branton, D. Three Decades of Nanopore Sequencing. *Nat. Biotechnol.* **34**, 518-524 (2016). DOI: 10.1038/nbt.3423.

Reviewer #2 (Remarks to the Author):

C1: In the manuscript “Targeted, high-resolution sensing of volatile organic compounds by covalent nanopore detection”, Lauren E M. and co-authors presents a groundbreaking approach to the sensing of aldehyde compounds through a covalent nanopore-based sensing platform, offering high-resolution detection with single-molecule manner. However, I think this method might lack of meaningful application and practical significance-then it might not worth publication on Nature Communications. Recent works, such as those reported in Trends in Analytical Chemistry (171, 117548, 2024) and Microchemical Journal (199, 110051, 2024), have demonstrated the analysis of aldehydes in volatile organic compounds (VOCs) and the environment with a limit of detection (LOD) as low as ~0.1 nM. This sensitivity is significantly higher than the current nanopore methods, which operate at the millimolar level. While the authors highlight the high-resolution sensing capabilities of their nanopore method for aldehyde isomers, such precision may currently hold limited practical value, especially when considering the cost-effectiveness and simplicity of alternative approaches, which is much more sensitive than the current nanopore methods (millimolar level). Although the author considered the nanopore methods featured high-resolution sensing for aldehydes isomers, such precision holds few practical values at the present concentration even in a single-molecule manner, particularly when considering the cost-effectiveness and simplicity of alternative approaches. Furthermore, the authors do not provide sufficient discussion on the scientific advancements and innovations that distinguish their work from previous efforts with similar goals. It would be helpful to understand the unique contributions of this study and why it warrants publication in a high-impact journal like Nature Communications.

A1: The reviewer has rightly pointed out that the current gold standard for the detection of volatile organic compounds is liquid or gas chromatography-mass spectrometry (LC/GC-MS). We note that pre-concentration steps are typically coupled to LC/GC-MS to achieve the limits of detection reported [Song, Microchem. J., 2024]. Additionally, a significant number of published LC/GC-MS approaches include a pre-derivatisation step to improve analyte detection and stability [El-Maghrabey, Trends Anal. Chem., 2024].

For point-of-care diagnostics, we believe that nanopore detection of VOCs will be better suited for the following reasons:

1. There is no requirement for pre-derivatisation (e.g., with hydrazine-based chromophores for aldehydes).
2. Devices for nanopore detection can be miniaturised and made portable (e.g., Oxford Nanopore's MinION [Dreameer, Nat. Biotechnol., 2016]).
3. Such devices are easy to operate and do not require specialised training or technical support staff.
4. The devices are inexpensive.

The present work provides a proof of concept for how nanopores can be used for VOC detection. Steps to improve the limits of detection will be explored in future studies and are discussed in the Conclusions. These include:

1. Solid-phase microextraction (SPME) can be used to pre-concentrate analytes in a large volume of exhaled breath into a smaller sample volume [Hamidi, *Crit. Rev. Anal. Chem.*, 2023]. Given that aldehydes ranging from propanal to nonanal are exhaled at 3-300 pM concentrations [Poli, *J. Chromatogr.*, 2010], 3-300 μ M concentrations of aldehydes can theoretically be obtained by concentrating 5 L of breath sample (which can be obtained in \sim 1 min [Pleil, *J. Breath Res.*, 2021]) into 5 μ L. Detection of analytes in nanolitre droplets (i.e., 250-350 nL) was previously demonstrated in a microfluidics set-up, and could equivalently be integrated here [Czekalska, *Lab Chip*, 2015].
2. Selective pre-concentration of aldehydes can be conducted by using SPME devices with fibre coatings selective for aldehydes [Poli, *J. Chromatogr.*, 2010; Yu, *Microchim. Acta*, 2018; Noreña-Caro, *Mater. Des.*, 2016] – this would simultaneously eliminate other potentially interfering chemical classes that might hinder disease diagnostics (see A3).
3. Nanopore engineering, e.g., nanopore mutants engineered with more than one covalent sensing site [Wu, *JACS*, 2008] or site-directed mutagenesis of proximate residues to improve cysteine thiol reactivity [Bo, *ACIE*, 2023] can give a >10 -fold improvement in limit of detection.
4. Nanopore sensing devices containing arrays of pores, e.g., the MinION device contains up to 512 active pores per flow cell [Dreamer, *Nat. Biotechnol.*, 2016]) can give a \sim 500-fold improvement in limit of detection.

In the present work, we have thoroughly covered the chemical aspects of aldehyde detection, as well as the use of protein engineering to improve the sensitivity of detection. We have made the following advances which have not been reported before:

1. Reversible hemithioacetal chemistry for the detection of small-molecule aldehydes in nanopores, for which dwell times in the range of milliseconds are essential for practical applications.
2. Nanopore measurements of rate constants for hemithioacetal adduct formation and dissociation, measurements of aldehyde hydration and aldehyde solubilities.
3. Rational protein engineering allowing the discrimination of diastereomeric hemithioacetal adducts, which aids the ability to distinguish closely related and isomeric aldehydes.
4. Expansion of the approach for the detection of mono alcohols with nanopores by enzyme-mediated conversion to aldehydes.

These achievements are reported in the Abstract, Introduction and Conclusions of the manuscript. We believe these contributions to the field of nanopore detection and small-molecule sensing meet with the high calibre of *Nature Communications*. In particular, they show the power and potential of targeted covalent nanopore detection.

C2: Additionally, the title mentions "targeted" sensing, but the authors do not clearly explain what this term entails. Clarification on the targeting mechanism and its specificity would be beneficial.

A2: We note that breath samples are chemically complex (i.e., >4000 VOCs, including alcohols, ketones, acids, hydrocarbons, thiols, ethers, esters). "Targeted" sensing refers to the use of reversible thiol-aldehyde chemistry to selectively detect aldehydes (which account for ~5 % of VOCs of human origin) amongst other chemical classes in the breath, reducing the complexity of small-molecule fingerprints, while retaining significant diagnostic value. The significance and importance of "targeted" sensing has been highlighted in the introduction (Page 3, lines 16-20).

Our discovery that reversible thiol-aldehyde chemistry can be used for small-molecule nanopore detection is novel. At pH 6.8, hemithioacetal adduct formation is frequent enough (e.g., ~500 events within 10 min for 3 mM butanal) and of sufficient lifetimes to allow for aldehyde identification from $I_{res\%}$ values, but not so long that the number of events is limited (e.g., ~130 ms for butanal). Other chemical classes would not be sufficiently reactive, and would therefore not have a long enough residence time in the nanopore to interfere with aldehyde detection.

We have demonstrated the "targeted" sensing of aldehydes in Fig. 5, wherein only the aldehydes in an aldehyde/alcohol mixture were detected. We have reiterated this point by including the following change in the Abstract:

"In this work, we show that covalent nanopore sensing can be used for the targeted detection of a reduced set of analytes in a mixture: in this case aldehydes, which constitute ~5% of human volatiles, can be selectively detected by using reversible thiol-aldehyde chemistry."

C3: In the authors' previous work, it was demonstrated that many types of small molecules can react with the T115C site inside the nanopore. However, the current study lacks an investigation into potential interference from other compounds, leading to insufficient discussion regarding the method's selectivity. A comprehensive analysis of potential interferences and strategies to mitigate them would strengthen the manuscript.

A3: The reviewer has raised a valid point. While most chemical classes would not have sufficient reactivity towards the single-cysteine nanopores used in this work (see A2), a few other potential analytes, such as a limited class of small-molecule disulfides will irreversibly react with the nanopore [Bo, ACIE, 2023].

Owing to the distinct differences in reactivity between disulfides and aldehydes (i.e., disulfides react with thiols irreversibly while aldehydes react reversibly), the two chemical classes are easily distinguishable. Nanopores that have reacted with

disulfides would not react further with aldehydes [Bo, ACIE, 2023], and these irreversibly modified nanopores would then be disregarded during an analysis with a flow cell containing multiple nanopores [Dreamer, Nat. Biotechnol., 2016]. Furthermore, pre-concentration steps with aldehyde-selective SPME devices would remove small-molecule disulfides from a sample mixture (see A1). A discussion is now included in Supplementary Section 3.

C4: Moreover, the fundamental for the formation of hemithioacetal underlying this work are not supported by sufficient experimental evidence, raising questions about the reliability and validity of the proposed approach. Mass or MRI evidences were suggested for the formation of hemithioacetal between the AG-T115C and aldehydes substrates in single molecular experimental conditions. The authors conducted rigorous experiments to rule out changes in the substrate concentration caused by aldehyde hydrolysis. In that case, can the formation of the hemithioacetal only occur via nucleophilic addition reaction? Could the nucleophilic substitution of the hydrolysis products also be a possible reaction pathway? The likelihood of this pathway might be related to the pH values.

A4: We clarify that hemithioacetal formation has been observed for the reaction between thiols and various aldehydes under aqueous conditions and at neutral pH (e.g., pH 6.5 – 7.2) using ^1H NMR [Bracchi, Chem. Commun., 2015; Caraballo, ACIE, 2010]. Additionally, we detected the hemithioacetal product formed from the reaction between hexanal and N-acetyl cysteine methyl ester at pH 6.8 with high resolution mass spectrometry (HRMS) (i.e., HRMS found $[\text{M}+\text{Na}]^+ = 300.1242$; $\text{C}_{12}\text{H}_{23}\text{NNaO}_4\text{S}$ requires 300.1240.) These have now been included in Supplementary Section 2.

We reiterate that aldehydes and their hydrates exist in equilibrium in aqueous solution. Hence, the concentration of free aldehyde present in solution is lower than that calculated from the mass of aldehyde added to a test recording. Further, hydration must be taken into account if the overall concentration of an analyte is to be determined from a current signal.

The reviewer has asked if hemithioacetal formation can occur by a direct nucleophilic substitution reaction (i.e., $\text{S}_{\text{N}}2$) between the cysteine thiolate and the diol hydration product (Fig. R1).

Fig. R1: Proposed nucleophilic substitution pathway by the reviewer.

At pH 6.8, hemithioacetal adduct formation is base-catalysed (i.e., higher rates of adduct formation were observed at pH values >6.8). This would disfavour the proposed nucleophilic substitution pathway given that protonation of the poor hydroxyl leaving group would be expected to participate. Additionally, stopped-flow experiments

conducted under base-catalysed conditions revealed that hemithioacetals are formed from aldehydes, and not the diol [Lienhard, JACS, 1966; this has been referenced in the manuscript (Ref 24)]. We therefore conclude that the proposed S_N2 pathway does not occur.

C5: The assertion that events A and B stem from reaction-formed enantiomers of the same aldehyde lacks robust experimental evidence. To strengthen this claim, the authors could provide definitive evidence linking the A/B event to the specific R/S configuration of the aldehyde enantiomers. This could open up an exciting new direction for further research and help solidify the understanding of the underlying mechanisms.

A5: Given the rapid reversible nature of hemithioacetal formation, it is not possible to isolate diastereomeric hemithioacetals in bulk for assignment of R/S configurations. To address the reviewer's concerns, we note the following experimental evidence to support our claim that the distinct levels A and B correspond to two potential diastereomers formed upon hemithioacetal formation. As some of these are unpublished results related to ongoing work, only Fig. R2 been included in the present manuscript.

1. The formation of the presumed diastereomeric adducts A and B followed bimolecular kinetics and adduct dissociation followed unimolecular kinetics (Fig. R2 and R3). Fig. R2 is now included as Supplementary Fig. 11.
2. More than 30 aldehydes that reacted within the (MK)₆(MK-T115C), (AG)₆(AG-G137C) and (AG)₆(AG-G137C-Ala3) nanopores produced reversible events with 2 distinct current levels. The tested aldehydes included aliphatic aldehydes, benzaldehydes and pyridinecarboxaldehydes.
3. Enantiomeric aldehydes (e.g., 2-(methylsulfinyl)benzaldehyde) containing one chiral centre gave four diastereomeric adducts with distinct current levels (Fig. R4).

Fig. R2: Kinetic analysis of the reaction of hexanal with $(MK)_6(MK-T115C)$. Left: Rates of adduct formation (v_{on}) and dissociation (v_{off}) for diastereomeric adducts A and B plotted against unhydrated hexanal concentration (determined from 1H NMR, Supplementary Section 4). Errors are standard deviations of rates at each concentration. Right: Single-channel recordings with 0.0, 0.5, 1.0, 1.5 and 2.0 mM unhydrated hexanal (trans). Current levels correspond to the two possible diastereomeric adducts, diastereomer A (2) and diastereomer B (3), and the unoccupied nanopore (1).

Fig. R3: Kinetic analysis of the reaction of benzaldehyde with $(AG)_6(AG-G137C)$. Left: Rates of adduct formation (v_{on}) and dissociation (v_{off}) for diastereomeric adducts A and B plotted against total benzaldehyde concentration. Errors are standard deviations of rates at

each concentration. Right: Single-channel recordings with 2.4, 4.5, 7.0 and 9.1 mM total benzaldehyde (trans). Current levels correspond to the two possible diastereomeric adducts, diastereomer A (2) and diastereomer B (3) and the unoccupied nanopore (1).

Fig. R4: Diastereomeric adducts formed by enantiomeric aldehydes reacting within $(AG)_6(AG-G137C-Ala3)$. **a**, Structures of 2-(methylsulfinyl)benzaldehyde (MSBenz) enantiomers. **b**, Percentage residual currents ($I_{res\%}$) calculated for each of the four current levels A, B, C, and D, which were assigned as four diastereomeric adducts derived from the reaction of MSBenz with $(AG)_6(AG-G137C-Ala3)$. Standard deviations of $I_{res\%}$ across different nanopores ($N = 3$ pores) are shown. **c**, Current trace showing the four distinct current blockades. Conditions: 5 mM MSBenz (trans), $(AG)_6(AG-G137C-Ala3)$ nanopore, 2 M KCl, 200 mM PIPES (pH 6.8), 20 μ M EDTA, recorded at -50 mV (trans), 21 ± 1 $^{\circ}$ C. Signals were low-pass filtered at 10 kHz and sampled at 50 kHz. Traces were filtered at 200 Hz for display.

C6: Additionally, in Figure 4d, the $I_{res\%}$ values for pentanal and hexanal display unexpected increases, which remain inadequately explained in the manuscript. Clarifying this anomaly would greatly enhance the comprehensiveness of the study.

A6: The unexpected increases were observed in the $(AG)_6(AG-G137C)$ nanopore. As the sensing region in the $(AG)_6(AG-G137C)$ nanopore is different from the other nanopores (i.e., the cysteine is at the 137 position, and different residues surround the cysteine by comparison with those around position 115), we would expect interactions between the hemithioacetal adducts with the $(AG)_6(AG-G137C)$ nanopore's sensing region to be different from the other nanopores. These distinct interactions must give rise to the unexpected trends in $I_{res\%}$. In this work, we made use of these unexpected interactions to differentiate between isomeric aldehydes (Fig. 4f).

C7: In the third paragraph of the introduction part, the author elaborated on the significance of rational detection of various VOC. However, this section lacks a direct connection with the subsequent experiments. This disconnection not only confuses the readers but undermines the logical coherence of this part. If these measurement methods are not further validated or applied in the experiments, then dedicating extensive space to discussing their significance in the preface appears redundant and inappropriate.

A7: We stated in paragraph 3 that a ratiometric fingerprint of aldehydes will inform on the health of a patient. We believe this to be an important consideration for the field of VOC detection and disease diagnosis, and have opted to keep it in the manuscript.

We have demonstrated ratiometric fingerprinting in Fig. 3, wherein pentanal/butanal mixtures of different ratios were quantified with good accuracy (i.e., <10% difference between expected and measured concentrations). A discussion is now included in the manuscript (page 10, lines 3-12) and is thoroughly discussed in Supplementary Section 6.

And some small questions:

C8: The chemical reaction equation in figure 1 seems confusion.

A8: We believe that the aldehyde hydration reaction in Fig. 1b should be included as it represents the state of the analyte in solution.

Aldehydes exist in equilibrium with their hydrates. Because the hydrate is unreactive, this reduces the concentration of free aldehyde that is able to react with the nanoreactor cysteine thiolate (see A4). This is explained in the manuscript (page 8, line 8 onwards) and is an important consideration for the quantitative detection of aldehydic VOCs.

C9: The figure 2 lacks the current of the open-pore state.

A9: We have included the open-pore state in Fig. 2a and all subsequent figures.

C10: The k_{off} for Benzaldehyde with the nanopore was much higher than other aldehydes, while its k_{on} is reasonable. Further interpretation is suggested of this kinetics data.

A10: The k_{off} for benzaldehyde is $38 \pm 2 \text{ s}^{-1}$ while the value is 5 to 10 s^{-1} for aliphatic aldehydes. The lower activation energy for hemithioacetal dissociation for benzaldehydes can be attributed to additional resonance stabilisation in the transition state conferred by conjugation of the incipient aldehyde group with the aromatic phenyl ring. A discussion is now included in Supplementary Section 3.

Additional measurements with the (AG)₆(AG-G137C-Ala3) nanopore also showed that benzaldehyde had higher k_{off} values compared to aliphatic aldehydes ($k_{\text{off}} = 6.4 \pm 0.1 \text{ s}^{-1}$ and $21 \pm 1 \text{ s}^{-1}$ for benzaldehyde (N = 3 pores), $0.82 \pm 0.07 \text{ s}^{-1}$ and $1.8 \pm 0.2 \text{ s}^{-1}$ for butanal (N = 3 pores), $0.86 \pm 0.07 \text{ s}^{-1}$ and $2.5 \pm 0.4 \text{ s}^{-1}$ for 2-methylpropanal (N = 3 pores)). These newly obtained rate constants are included in Supplementary Table 6.

C11: The k_{off} of isomer B was generally higher than that of its isomer A. If the two signals indeed arise from the chirality induced post-addition, it raises a compelling question as to why the dissociation rates of the addition products with different chirality exhibit such significant disparities.

A11: The protein nanopores are chiral as they are composed of *L*-amino acids. Prochiral aldehydes will react with the cysteine thiolate to give diastereomeric hemithioacetal adducts. Differences in the diastereomer interactions with the chiral protein wall are expected to give rise to differences in rate constants of adduct formation and dissociation, as well as differences in $I_{\text{res}\%}$. This was previously observed for the reversible formation of organoarsenic (III) adducts within a different chiral protein nanopore [Shin, *ACIE*, 2007].

The lower $I_{\text{res}\%}$ of adduct B suggests that adduct B is on average further from the protein nanopore surface than adduct A. Accordingly, adduct B is hypothesised to experience weaker adduct-nanopore interactions than adduct A, and undergo faster dissociation.

References:

[1] Song, J., Li, R., Yu, R., Zhu, Q., Li, C., He, W. & Liu, J. Detection of VOCs in Exhaled Breath for Lung Cancer Diagnosis. *Microchem. J.* **199**, 110051, (2024). DOI: 10.1016/j.microc.2024.110051.

[2] El-Maghrabey, M. H., Hashem, H. M., El Hamd, M. A., El-Shaheny, R., Kishikawa, N., Kuroda, N. & Magdy, G. Comprehensive Greenness Evaluation of the Reported Chromatographic Methods for Aldehydes Determination as Clinical Biomarkers and Food Quality Indicators. *Trends Anal. Chem.* **171**, 117548 (2024). DOI: 10.1016/j.trac.2024.117548.

[3] Dreamer, D., Akeson, M. & Branton, D. Three Decades of Nanopore Sequencing. *Nat. Biotechnol.* **34**, 518-524 (2016). DOI: 10.1038/nbt.3423.

[4] Hamidi, S. Recent Advances in Solid-Phase Extraction as a Platform for Sample Preparation in Biomarker Assay. *Crit. Rev. Anal. Chem.* **53**, 199-210 (2023). DOI: 10.1080/10408347.2021.1947771.

[5] Poli, D., Goldoni, M., Corradi, M., Acampa, O., Carbognani, P., Internullo, E., Casalini, A. & Mutti, A. Determination of Aldehydes in Exhaled Breath of Patients with Lung Cancer by Means of On-Fiber-Derivatisation SPME–GC/MS. *J. Chromatogr. B*, **878**, 2643-2651 (2010). DOI: 10.1016/j.jchromb.2010.01.022.

- [6] Pleil, J. D., Wallace, M. A. G., Davis, M. D. & Matty, C. M. The Physics of Human Breathing: Flow, Timing, Volume, and Pressure Parameters for Normal, On-demand, and Ventilator Respiration. *J. Breath Res.* **15**, 042002 (2021). DOI: 10.1088/1752-7163/ac2589.
- [7] Czekalska, M. A., Kaminski, T. S., Jakiela, S., Sapra, K. T., Bayley, H. & Garstecki, P. A Droplet Microfluidic System for Sequential Generation of Lipid Bilayers and Transmembrane Electrical Recordings. *Lab Chip* **15**, 541-548 (2015). DOI: 10.1039/c4lc00985a.
- [8] Yu, L.-Q., Wang, L.-Y., Su, F.-H., Hao, P.-Y., Wang, H. & Lv, Y.-K. A Gate-Opening Controlled Metal-Organic Framework for Selective Solid-Phase Microextraction of Aldehydes from Exhaled Breath of Lung Cancer Patients. *Microchim. Acta* **185**, 307 (2018). DOI: 10.1007/s00604-018-2843-1.
- [9] Noreña-Caro, D. & Álvarez-Láinez, M. Functionalization of Polyacrylonitrile Nanofibers with β -Cyclodextrin for the Capture of Formaldehyde. *Mater. Des.* **95**, 632-640 (2016). DOI: 10.1016/j.matdes.2016.01.106.
- [10] Wu, H.-C. & Bayley, H. Single-Molecule Detection of Nitrogen Mustards by Covalent Reaction within a Protein Nanopore. *J. Am. Chem. Soc.* **130**, 6813-6819 (2008). DOI: 10.1021/ja8004607.
- [11] Bo, Z., Lim, Z. H., Duarte, F., Bayley, H. & Qing, Y. Mobile Molecules: Reactivity Profiling Guides Faster Movement on a Cysteine Track. *Angew. Chem. Int. Ed.* **62**, e2023008 (2023). DOI: 10.1002/anie.202300890.
- [12] Bracchi, M. E. & Fulton, D. A. Orthogonal Breaking and Forming of Dynamic Covalent Imine and Disulfide Bonds in Aqueous Solution. *Chem. Commun.* **51**, 11052-11055 (2015). DOI: 10.1039/c5cc02716k.
- [13] Caraballo, R., Dong, H., Ribeiro, J. P., Jiménez-Barber, J. & Ramström, O. Direct STD NMR Identification of β -Galactosidase Inhibitors from a Virtual Dynamic Hemithioacetal System. *Angew. Chem. Int. Ed.* **49**, 589-593 (2010). DOI: 10.1002/anie.200903920.
- [14] Lienhard, G. E. & Jencks, W. P. Thiol Addition to the Carbonyl Group. Equilibria and Kinetics. *J. Am. Chem. Soc.* **88**, 3982-3995 (1966). DOI: 10.1021/ja00969a017.
- [15] Shin, S.-H., Steffensen, M. B., Claridge, T. D. W. & Bayley, H. Formation of a Chiral Center and Pyrimidal Inversion at the Single-Molecule Level. *Angew. Chem. Int. Ed.* **46**, 7412-7416 (2007). DOI: 10.1002/anie.200700736.

Reviewer #3 (Remarks to the Author):

With great interest I read the new manuscript by H. Bayley and coworkers. The team is among the leading one in the area of nanopore sensing having an enormous expertise in using hemolysine as molecular sensor. Here they presents novel ways to detect small aldehydes by nanopores. Sensing is achieved by blocking the ion current in a specific manner to allow discrimination of a larger number of different aldehydes. For this the team introduced a cysteine next to the constriction zone. If an aldehyde passes the cysteine they will react and bind temporarily and modulate eventually the ion current. Further enhancement in selectivity was done by additional mutations giving a certain cooperativity. Theoretically this is obvious in the nanopore field, however, finding an appropriate binding site and causing specific ion current modulation is a real challenge. Able to discriminate different aldehydes is a rather unique finding.

I think this is a beautiful example for enzyme engineering in a rational manner with single molecule resolution. With the current goal of bringing nanopore detection to the market this approach would be a possible further application.

We thank the reviewer for their positive comments on the work presented in this manuscript.

C1: However, I see a few bottlenecks not mentioned in the otherwise very clearly written introduction: how to bring volatile molecules into an aqueous solution knowing that their solubility is very different or even very low for various aldehydes (some have even very low solubility). This is particular important as the ratio seems to matter often. Maybe using water/ethanol solutions and solid state nanopores? To my knowledge nobody has tried that).

A1: The reviewer raises the question of how to bring volatile organic compounds into aqueous solution, particularly for aldehydes with low solubilities. We have included the following discussion points in the Conclusions of the manuscript:

1. Solid-phase microextraction (SPME) can be used to pre-concentrate analytes in a large volume of exhaled breath into a smaller sample volume [Hamidi, Crit. Rev. Anal. Chem., 2023]. Given that aldehydes ranging from propanal to nonanal are exhaled at 3-300 pM concentrations [Poli, J. Chromatogr., 2010], 3-300 μ M concentrations of aldehydes can theoretically be obtained by concentrating 5 L of breath sample (which can be obtained in \sim 1 min [Pleil, J. Breath Res., 2021]) into 5 μ L. Detection of analytes in nanolitre droplets (i.e., 250-350 nL) was previously demonstrated in a microfluidics set-up, and could equivalently be integrated here [Czekalska, Lab Chip, 2015].
2. Selective pre-concentration of aldehydes can be conducted by using SPME devices with fibre coatings selective for aldehydes [Poli, J. Chromatogr., 2010; Yu, Microchim. Acta, 2018; Noreña-Caro, Mater. Des., 2016] – this would

simultaneously eliminate other potentially interfering chemical classes that might hinder disease diagnostics.

3. Aldehydic VOCs might be concentrated into water/ethanol solutions to overcome solubility issues for longer chain aldehydes. However, this would require an exploration of alternative membranes with improved tolerance to ethanol (e.g., block copolymer "bilayers"), a worthwhile future endeavour.

The functionalisation of solid state nanopores with a single thiol group or the discrimination of small molecules differing by a single $-CH_2-$ with solid-state nanopores have not been achieved. Therefore, non-protein pores have been excluded from our discussions.

C2: The second bottleneck is the different reaction rate already mentioned in the manuscript.

Would adding higher PEG concentration help to dehydrate the aldehydes?

A2: Aldehyde hydration leads to a decrease in the concentration of free aldehyde. PEG might reduce the hydration of aldehydes by altering the activity of water, but we have been unable to find any experimental literature to support this. Further, reactions between impurities commonly found in PEG can form low-molecular-weight aldehydes, which may interfere with our detection method [Hemenway, J. Pharm. Sci., 2012].

However, alternative means of increasing the limit of detection might be useful and have been discussed in the Conclusions of the manuscript. They include:

1. Pre-concentration of aldehydes from large volumes of breath will increase the concentration of aldehydes for detection (see A1).
2. Nanopore engineering: e.g., nanopore mutants with more than one covalent sensing site [Wu, JACS, 2008] or mutagenesis of proximate residues to improve cysteine thiol reactivity [Bo, ACIE, 2023]).
3. Nanopore sensing devices containing arrays of pores (e.g., the MinION device contains up to 512 nanopores [Dreamer, Nat. Biotechnol., 2016]).

C3: You use lipid membranes, what about partitioning of hydrophobic aldehydes into the membrane reducing the concentration in a specific manner?

A3: We clarify that under our experimental conditions, we did not observe any significant reductions in hemithioacetal adduct formation rates (v_{on}) over long recording times. For example, rates of hemithioacetal adduct formation obtained for hexanal were generally consistent across 2-minute intervals in single-channel recordings (**Fig. R1**).

Fig. R1: Rates of adduct formation calculated from 2-minute intervals. Rates of adduct formation (v_{on}) calculated from 2-minute intervals were consistent for hexanal in the (AG)₆(AG-T115C) nanopore. Recordings with hexanal were conducted for <7 minutes. Errors are standard deviations of rates at each concentration.

We note that the volume of the lipid bilayer formed in our set-up is very small (i.e., $3.45 \times 10^{-14} \text{ dm}^3$, calculated using an approximate lipid bilayer radius of $50 \text{ }\mu\text{m}$ [Bayley, 2008] and a bilayer thickness of 4.4 nm [Chen, Nano Lett., 2023]) compared to the volume of solution used (i.e., $5 \times 10^{-4} \text{ dm}^3 = 500 \text{ }\mu\text{L}$). Using a partition coefficient of 50 [Ramsden, Experientia, 1993], approximately $0.17 \text{ }\mu\text{g}$ of hexanal would have partitioned into the lipid bilayer if $50 \text{ }\mu\text{g}$ of hexanal (i.e., 1 mM hexanal) were added under our experimental setup. Hence, loss of aldehydes from partitioning into the bilayer would minimally affect the rates of adduct formation.

Nevertheless, we note that this could be a potential issue when transferring this technology to commercial devices which use block copolymer bilayers. This could be addressed by screening for suitable polymers which aldehydes minimally partition into, or by characterising the rate and extent of analyte partitioning (i.e., for subsequent error correction).

The work is complete and fully described. To reproduce is technically a challenge but feasible for competing groups. The manuscript is sound and at the current edge in the nanopore field.

Overall this work is a significant step and I recommend publication.

References:

- [1] Hamidi, S. Recent Advances in Solid-Phase Extraction as a Platform for Sample Preparation in Biomarker Assay. *Crit. Rev. Anal. Chem.* **53**, 199-210 (2023). DOI: 10.1080/10408347.2021.1947771.

- [2] Poli, D., Goldoni, M., Corradi, M., Acampa, O., Carbognani, P., Internullo, E., Casalini, A. & Mutti, A. Determination of Aldehydes in Exhaled Breath of Patients with Lung Cancer by Means of On-Fiber-Derivatisation SPME–GC/MS. *J. Chromatogr. B*, **878**, 2643-2651 (2010). DOI: 10.1016/j.jchromb.2010.01.022.
- [3] Pleil, J. D., Wallace, M. A. G., Davis, M. D. & Matty, C. M. The Physics of Human Breathing: Flow, Timing, Volume, and Pressure Parameters for Normal, On-demand, and Ventilator Respiration. *J. Breath Res.* **15**, 042002 (2021). DOI: 10.1088/1752-7163/ac2589.
- [4] Czekalska, M. A., Kaminski, T. S., Jakiela, S., Sapra, K. T., Bayley, H. & Garstecki, P. A Droplet Microfluidic System for Sequential Generation of Lipid Bilayers and Transmembrane Electrical Recordings. *Lab Chip* **15**, 541-548 (2015). DOI: 10.1039/c4lc00985a.
- [5] Yu, L.-Q., Wang, L.-Y., Su, F.-H., Hao, P.-Y., Wang, H. & Lv, Y.-K. A Gate-Opening Controlled Metal-Organic Framework for Selective Solid-Phase Microextraction of Aldehydes from Exhaled Breath of Lung Cancer Patients. *Microchim. Acta* **185**, 307 (2018). DOI: 10.1007/s00604-018-2843-1.
- [6] Noreña-Caro, D. & Álvarez-Láinez, M. Functionalization of Polyacrylonitrile Nanofibers with β -Cyclodextrin for the Capture of Formaldehyde. *Mater. Des.* **95**, 632-640 (2016). DOI: 10.1016/j.matdes.2016.01.106.
- [7] Hemenway, J. N., Carvalho, T. C., Rao, V. M., Wu, Y., Levons, J. K., Narang, A. S., Paruchuri, S. R., Stamato, H. J. & Varia, S. A. Formation of Reactive Impurities in Aqueous and Neat Polyethylene Glycol 400 and Effects of Antioxidants and Oxidation Inducers. *J. Pharm. Sci.* **101**, 3305-3318 (2012). DOI: 10.1002/jps.23198.
- [8] Wu, H.-C. & Bayley, H. Single-Molecule Detection of Nitrogen Mustards by Covalent Reaction within a Protein Nanopore. *J. Am. Chem. Soc.* **130**, 6813-6819 (2008). DOI: 10.1021/ja8004607.
- [9] Bo, Z., Lim, Z. H., Duarte, F., Bayley, H. & Qing, Y. Mobile Molecules: Reactivity Profiling Guides Faster Movement on a Cysteine Track. *Angew. Chem. Int. Ed.* **62**, e2023008 (2023). DOI: 10.1002/anie.202300890.
- [10] Dreamer, D., Akeson, M. & Branton, D. Three Decades of Nanopore Sequencing. *Nat. Biotechnol.* **34**, 518-524 (2016). DOI: 10.1038/nbt.3423.
- [11] Bayley, H., Luchian, T., Shin, S. H. & Steffensen, M. B., in *Single Molecules and Nanotechnology*, Springer Berlin Heidelberg, **2008**, pp. 251-277.
- [12] Chen, T., Ghosh, A. & Enderlein, J. Cholesterol-Induced Nanoscale Variations in the Thickness of Phospholipid Membranes. *Nano Lett.* **23**, 2421-2426 (2023). DOI: 10.1021/acs.nanolett.2c04635.
- [13] Ramsden, J. J. Partition Coefficients of Drugs in Bilayer Lipid Membranes. *Experientia* **49**, 688-692 (1993). DOI: 10.1007/BF01923952.

RESPONSE TO REVIEWERS

Reviewer #1 (Remarks to the Author):

C1: The authors fully improved the manuscript and addressed my questions satisfactorily.

I think that the manuscript is ready for publication in Nature Communication.

A1: We thank the reviewer for their time and effort spent in improving the manuscript.

Reviewer #2 (Remarks to the Author):

C1: While the authors have made certain revisions, the fundamental concerns regarding the scientific and practical validity of the proposed method remain unresolved. The core claims are not sufficiently supported by data, and the potential for practical application is, at best, remote. I do not recommend this work for publication in its current form.

A1: We argue that our work is scientifically rigorous and has practical importance. In this work, we have detailed a new chemistry for the targeted detection of aldehydes using nanopores. We have unambiguously demonstrated single-molecule discrimination of structurally similar aldehydes. This is accompanied by detailed characterisation of reaction kinetics, analyte solubilities, and effective analyte concentrations due to hydration with practical applications in mind. This sets the groundwork for future diagnostics based on VOC detection.

C2: Firstly, the claim that this approach could be applied to point-of-care testing (POCT) for VOCs remains speculative. Although the authors reference various strategies such as preconcentration, multi-channel detection, and nanopore engineering, these are drawn from unrelated contexts and are not convincingly integrated into their own method. The manuscript fails to provide any compelling evidence that the proposed system is truly low-cost, rapid, or user-friendly. In fact, critical barriers—including chip degradation in the MinION platform and the extensive computational resources required for nanopore signal processing—are entirely overlooked. Without addressing these limitations, the method cannot be considered viable for practical applications.

A2: We respectfully disagree with the assertion that our discussion of potential applications is speculative or unsupported. This work establishes hemithioacetal chemistry as a new and robust chemistry for targeted, single-molecule aldehyde detection using nanopores, supported by detailed profiling of sensing kinetics, solubilities and hydration constants for 11 aldehydes, and rational nanopore engineering for high-resolution aldehyde discrimination. These are advances with clear relevance for future sensing application.

While practical deployment is beyond the immediate goal of this paper, we note that the suggested barriers, such as MinION chip degradation, cost, speed, and data processing, are overstated. The MinION platform is widely recognised for being portable, low-cost, and user-friendly, and has already seen field deployment in genomics.

Towards building a deployable device, various well-established techniques have been suggested. These include sample preconcentration (used in numerous analytical technologies and within references the reviewer had previously suggested [Song, *Microchem. J.*, 2024]), multi-channel detection (used in the MinION and elsewhere) and nanopore engineering (used with effect in this paper).

We reiterate that the focus of the work remains on introducing and validating hemithioacetal chemistry as a new means for single-molecule aldehyde detection using nanopores, which lays the essential groundwork for future device development.

C3: Secondly, the authors continue to ignore the crucial role of the protein microenvironment in modulating cysteine reactivity, a fact well established in the literature and even acknowledged in their prior work. The supporting data—primarily derived from simplified chemical models—bear little relevance to the biological setting in which the claimed sensing would occur. Moreover, there is still no investigation into whether the monomers forming the nanopore might themselves participate in the reaction, calling into question the biological plausibility and specificity of the approach.

A3: The reviewer raises three points, which we address below.

First, the reviewer claimed that we have “continued to ignore the crucial role of the protein microenvironment in modulating cysteine reactivity”. While cysteine reactivity can influence detection frequency (page 14, lines 8-10), it does not affect the signal features critical for aldehyde discrimination. In fact, the position (thus the microenvironment) of cysteine in modulating adduct discrimination (and secondarily, cysteine reactivity) has been well documented in our previous work [Bo, *Angew. Chem. Int. Ed.*, 2023] and is again evident in Figure 4 of the current manuscript.

Second, the reviewer claimed that our work “bear little relevance to the biological setting in which the claimed sensing would occur”. We emphasise that the current work aims to establish hemithioacetal chemistry as a new means for nanopore sensing of aldehydes, a key step towards future applications. We do not claim direct aldehyde detection in biological settings, nor is this the intended scope of the work. Available techniques to enable such applications are discussed in the main text and in A2.

Third, the reviewer questions if the monomers forming the nanopore “might themselves participate in the reaction”. The α HL nanopores were isolated as SDS stable heptamers (see Experimental methods) which do not dissociate into monomers during single-channel recordings [Braha, *Chem. Biol.*, 1997]. No monomers were added during single-channel recordings.

C4: Thirdly, the mechanistic assumptions presented remain weak. The formation of hemiacetals is known to proceed via both acid- and base-catalyzed pathways, yet the authors make no attempt to experimentally assess pH dependence. Instead, they selectively cite literature to justify a nucleophilic addition mechanism, without conducting even a minimal set of control experiments. This lack of mechanistic rigor severely undermines the reliability of the proposed sensing strategy, particularly if it is to be applied in complex or variable environments.

A4: We clarify that hemithioacetal adducts are being formed in this work, not hemiacetals. Hemithioacetal formation can be acid- or base-promoted. In our system (pH 6.8), hemithioacetal formation is base-promoted. Specifically, we observed higher rates of adduct formation at pH values >6.8 and lower rates of adduct formation at pH values <6.8.

The reviewer suggests that hemithioacetals could be formed from nucleophilic substitution of diols (i.e., hydrated aldehyde). However, this was already disproved in reference 24 of the manuscript [Lienhard, *J. Am. Chem. Soc.*, 1966], which used stopped-flow kinetics to demonstrate that hemithioacetals formed from aldehydes rather than diols. The diol was also treated as an unreactive component during hemithioacetal formation in [Gunshore, *Bioorg. Chem.*, 1985], [Schwartz, *J. Org. Chem.*, 1996] and [Caraballo, *Angew. Chem. Int. Ed.*, 2010]. There is to our knowledge no credible literature to support the reviewer's proposed mechanism.

References:

- [1] Song, J., Li, R., Yu, R., Zhu, Q., Li, C., He, W. & Liu, J. Detection of VOCs in Exhaled Breath for Lung Cancer Diagnosis. *Microchem. J.* **199**, 110051, (2024). DOI: 10.1016/j.microc.2024.110051.
- [2] Bo, Z., Lim, Z. H., Duarte, F., Bayley, H. & Qing, Y. Mobile Molecules: Reactivity Profiling Guides Faster Movement on a Cysteine Track. *Angew. Chem. Int. Ed.* **62**, e2023008 (2023). DOI: 10.1002/anie.202300890.
- [3] Braha, O. et al. Designed protein pores as components for biosensors. *Chem. Biol.* **4**, 497–505 (1997). DOI: 10.1016/S1074-5521(97)90321-5.
- [4] Lienhard, G. E. & Jencks, W. P. Thiol Addition to the Carbonyl Group. Equilibria and Kinetics. *J. Am. Chem. Soc.* **88**, 3982-3995 (1966). DOI: 10.1021/ja00969a017.
- [5] Gunshore, S., Brush, E. J. & Hamilton, G. A. Equilibrium constants for the formation of glyoxylate thiohemiacetals and kinetic constants for their oxidation by O₂ catalyzed by L-hydroxy acid oxidase. *Bioorg. Chem.* **13**, 1–13 (1985). DOI: 10.1016/0045-2068(85)90002-1.
- [6] Schwartz, B., Vogel, K. W. & Drueckhammer, D. G. Coenzyme A Hemithioacetals as Easily Prepared Inhibitors of CoA Ester-Utilizing Enzymes. *J. Org. Chem.* **61**, 9356–9361 (1996). DOI: 10.1021/jo9616724.

[7] Caraballo, R., Dong, H., Ribeiro, J. P., Jiménez-Barbero, J. & Ramström, O. Direct STD NMR identification of β -galactosidase inhibitors from a virtual dynamic hemithioacetal system. *Angew. Chem. Int. Ed.* **49**, 589–593 (2010). DOI: 10.1002/anie.200903920.

Reviewer #3 (Remarks to the Author):

C1: I read the ms. as well as the reply. As stated in my first report the manuscript contains a novel approach which will raise sufficient interest in the community. The authors improved the ms. and it can be published as it is

A1: We thank the reviewer for their time and effort spent in improving the manuscript.